# Dominant Shuffle: An Incredibly Simple but Exceptionally Effective Data Augmentation Method for Time-Series Prediction

## Abstract

Frequency-domain data augmentation (DA) has shown strong performance in time-series prediction due to its ability to preserve data-label consistency. However, we observed that existing frequency-domain augmentations introduce excessive variability, leading to out-of-distribution samples that may be harmful to model performance. To address this, we introduced two simple modifications to frequency-domain DA. First, we limit perturbations to dominant frequencies with larger magnitudes, which capture the main periodicities and trends of the signal. Second, instead of using complicated random perturbations, we simply shuffle the dominant frequency components, which preserves the original structure while avoiding external noise. With the two simple modifications, we proposed dominant shuffle—a simple yet highly effective data augmentation technique for time-series prediction. Our method is remarkably simple, requiring only a few lines of code, yet exceptionally effective, consistently and significantly improving model performance. Extensive experiments on short-term, long term, few-shot and cold-start prediction tasks with eight state-of-the-art models, nine existing augmentation methods and twelve datasets demonstrate that dominant shuffle consistently boosts model performance with substantial gains, outperforming existing augmentation techniques. Our method is simple, practical, and effective. The code is available at `https://anonymous.4open.science/r/dominant-shuffle-A70E`.

## 1 Introduction

Time-series prediction aims to forecast multivariate future values based on historical observations. It is a long-standing problem with various applications in electricity pricing, weather forecast, traffic prediction Lim & Zohren (2021); Zhou et al. (2021). Recently, impressive results have been achieved by using various deep learning architectures, e.g. recurrent neural networks (RNNs) Rangapuram et al. (2018); Salinas et al. (2020); Ma et al. (2020), Transformers Zhou et al. (2021); Wu et al. (2021); Zhou et al. (2022b); Liu et al. (2024), and temporal convolutional networks (TCNs) Wang et al. (2023); Liu et al. (2022); Wu et al. (2023). Neural networks require a large volume of training data to effectively fit their numerous parameters. Unfortunately, time-series data acquired from real-world sensors are often limited in many time-series applications. The patterns of the time series heavily depend on specific dynamic system that generates the data and other data sources are not applicable Chen et al. (2023a); Semenoglou et al. (2023).

To mitigate the impact of insufficient data in time series analysis, several data augmentation techniques have been explored. Most of these data augmentation techniques in time series analysis focus on classification Qian et al. (2022); Um et al. (2017); Le Guennec et al. (2016); Steven Eyobu & Han (2018); Nam et al. (2020); Lim & Zohren (2021); Zhang et al. (2022b); Chen et al. (2023b) and anomaly detection Lim et al. (2018); Lim & Zohren (2021); Gao et al. (2020). The data-label coherence is a key factor to effective data augmentation Wen et al. (2021); Zhang et al. (2023); Sun et al. (2023). It measures the semantic connection between the augmented data and the label. These augmentations designed for classification modify only the input time series (data) without affecting the class labels, ensuring data-label coherence as long as the perturbed sample remains within the original category. However, the prediction task requires more fine-grained historical temporal variation to accurately estimate future dynamics Zhang et al. (2023); Chen et al. (2023a). Only

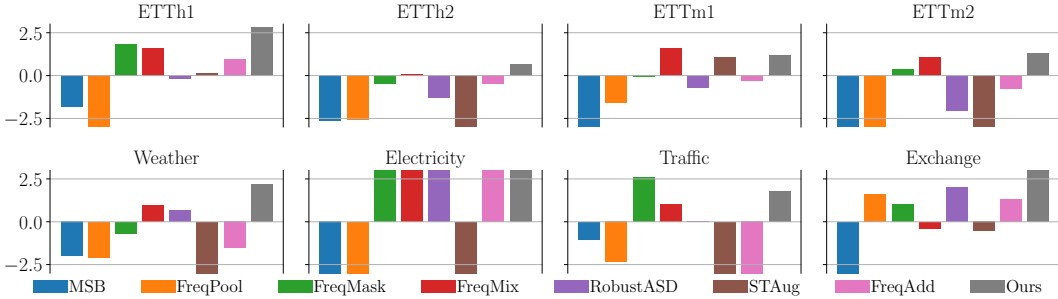

Figure 1: Relative improvements (%) of various data augmentations over the baseline on eight datasets using the state-of-the-art iTransformer Liu et al. (2024) model. Zero corresponds to the original model without any data augmentation. Our method consistently improves the baseline on all the datasets and outperforms other augmentations in most cases. The improvements are based on the average performance of four prediction lengths: 96, 192, 336, and 720.

perturbing the data could disrupt the data-label coherence and lead to performance degradation Zhang et al. (2023); Chen et al. (2023a).

Recently, to improve the data-label coherence for time-series prediction, Chen et al. (2023a) proposed to perturb the combined data (historical sequence) and labels (future sequences) in the frequency domain. This method first merges the data and label into a single sequence, then applies random perturbations in frequency domain, followed by conversion back to the time domain. However, existing frequency-domain augmentations apply random perturbations across the full spectrum[1], which can lead to excessive changes and produce samples that fall outside the original data distribution. Incorporating these out-of-distribution (OOD) samples into the training set creates a domain gap between the training and testing sets, which negatively affects model performance, particularly when a large number of augmented samples are used. In addition, random perturbations such as FreqMix, FreqAdd Zhang et al. (2022b) could introduce external noise and further enlarge the original-augmented gap.

In this paper, to reduce the domain gap between the augmented and original data, we propose to limit the perturbation in data augmentation. First, we restrict the perturbation to specific frequencies instead of full-spectrum perturbation. Several recent studies have pointed out that a few frequency components are dominating the periodicity and main trends of the time series, and other Frequencies correspond to minor trends or noise Wu et al. (2023); Zhou et al. (2022b;a). Following Wu et al. (2023), we perturb top-$k$ frequencies with highest magnitudes. Second, to avoid excess external noise, we use random shuffle for perturbation. Shuffle rearranges existing components without introducing any external randomness.

Extensive comparisons were made among eight state-of-the-art (SOTA) time series models, nine different data augmentation methods on eight public datasets using. These comparisons demonstrate that, despite its simplicity, our method significantly outperforms other competitors by a substantial margin. As shown in Fig. 1, our method consistently improves the performance of iTransformer Liu et al. (2024) model across various datasets, and outperforms other DA methods such as MSB Bandara et al. (2021), FreqPool Chen et al. (2023b), FreqMask Chen et al. (2023a), FreqMix Chen et al. (2023a), RobustASD Gao et al. (2020), and STAug Zhang et al. (2023) in most cases.

Comprehensive ablation studies demonstrate that perturbing dominant frequencies yields significantly better performance than various full-spectrum perturbations. And shuffle is proven to be superior to other randomization techniques. Besides, our augmentation demonstrates improved augmented-original gap over other augmentations, confirmed by both qualitative visualizations and quantitative results.

---

[1]By "full-spectrum perturbation" we mean that the perturbation could be possibly, not necessarily, applied to any frequency components.

## 2 RELATED WORK

In the last decade, deep learning has emerged as a powerful tool in time-series prediction and has shown superior performance over traditional statistical methods such as ARIMA and Exponential Smoothing McKenzie (1984). A rich line of studies has introduced various deep-learning architectures, including recurrent neural networks (RNNs) Rangapuram et al. (2018); Salinas et al. (2020); Ma et al. (2020), temporal convolution neural networks (TCNs) Wang et al. (2023); Liu et al. (2022); Wu et al. (2023), and Transformers Wu et al. (2021); Ni et al. (2023); Nie et al. (2023a); Liu et al. (2024); Zhou et al. (2022b). These models learn to predict the future from large volumes of historical data.

Various data augmentations have been proposed for time series data and many of these techniques were proposed for the classification tasks Wen et al. (2021); Qian et al. (2022); Um et al. (2017); Le Guennec et al. (2016); Steven Eyobu & Han (2018); Nam et al. (2020); Lim & Zohren (2021); Zhang et al. (2022b); Chen et al. (2023b). Many of these methods regard time series data as one-dimensional image and borrowed data augmentations, e.g. cropping Le Guennec et al. (2016); Cui et al. (2016) flipping Wen et al. (2021), and noise injection Wen & Keyes (2019), from computer vision. Window warping Wen et al. (2021) is a time series-specific data augmentation that upsamples (or downsamples) a random range of the time series while keeping other time ranges unchanged.

In addition to time-domain augmentations, there are also methods that perturb the original data in the frequency domain. Gao Gao et al. (2020) proposed to add noise on both magnitude and phase in the frequency domain. Zhang Zhang et al. (2022b) proposed to add single or multiple frequency components in the first half of the frequency spectrum. Chen Chen et al. (2023b) proposed to perform pooling or smoothing operations in the frequency domain.

While most of the augmentations focus on the classification tasks, a few methods for forecasting task have also been explored. Bandara Bandara et al. (2021) introduces two DA methods for forecasting : (i) Average selected with distance (ASD), which generates augmented time series using the weighted sum of multiple time series, and the weights are determined by the dynamic time warping (DTW) distance Forestier et al. (2017); (ii) Moving block bootstrapping (MBB) generates augmented data by manipulating the residual part of the time series after STL Decomposition Semenoglou et al. (2023) and recombining it with the other series. Zhang Zhang et al. (2023) proposed to simultaneously augment in frequency and time domains. Recently, Chen et. al. Chen et al. (2023a) proposed to augment both the data (historical sequence) and the label (future sequence) in the frequency domain to improve the data-label coherence. Although this method generally achieves decent results, full-spectrum randomization imposes a large domain gap between the augmented and the original data, sometimes leading to degraded performance.

## 3 DOMINANT FREQUENCY SHUFFLE FOR TIME-SERIES

### 3.1 TIME-SERIES PREDICTION AND FREQUENCY DOMAIN AUGMENTATION

Time-series prediction is a sequence-to-sequence problem where the model estimates a future multivariate sequence based on a sequence of historical measurements. Let $x = \{x^1, x^2, ..., x^L\}_{t=1}^{L} \in \mathbb{R}^{L \times D}$ be the $D$-dimensional historical sequence, and $y = \{x^{L+1}, x^{L+2}, ..., x^{L+T}\}_{t=L+1}^{L+T} \in \mathbb{R}^{T \times D}$ is the future sequence to be estimated. $x^t$ is the measurement at timestep $t$ and $D$ is the number of variates. Next, we will use $x \in R^{L \times D}$ and $y \in R^{T \times D}$ to denote the historical and future sequences. $x$ and $y$ are the input and output of time-series prediction models, respectively.

Deep neural networks learn the $x \to y$ mapping from large volume of $(x, y)$ pairs. DA expands the training set by adding perturbations to existing pairs. Ensuring data-label coherence hinges on applying consistent perturbations to both the data and label. Chen et al. (2023a) proposed to merge historical and future sequences, and then perturb the merged sequence in the frequency domain. The augmentation can be formulated as:

$$[\hat{x}, \hat{y}] = \text{iFFT}\Big(\text{perturb}\big(\text{FFT}([x, y])\big)\Big), \tag{1}$$

where FFT and iFFT are fast Fourier transform and the inverse fast Fourier transform, $[x, y]$ denotes the concatenation of two sequences. 'perturb' denotes the perturbation in the frequency domain, which can be arbitary perturbations such as frequency mix (FreqMix), mask (FreqMask) Chen et al. (2023a), and frequency add Zhang et al. (2022b).

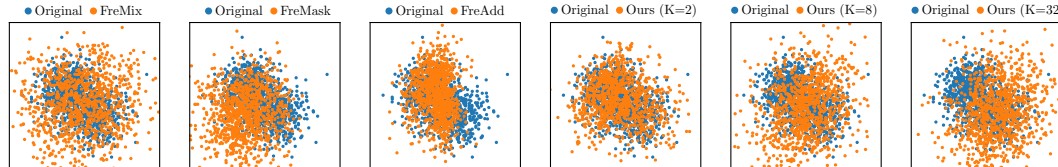

Figure 2: t-SNE visualization of features of various DA methods. An iTransformer model was trained with **original samples** on the ETTh1 (predict-96) dataset, and the features of both the original and different augmented samples were visualized using t-SNE. FreMix, FreMask, and FreAdd generate OOD samples. Our method with $K = 2$ aligns well with the original data, and increasing $K$ results in larger augmented-original gaps.

### 3.2 Dominant Frequency Shuffle

Existing frequency-domain augmentations like FreqMix and FreqMask Chen et al. (2023a) may perturb arbitrary frequency components, potentially causing significant deviations from the original data and introducing out-of-distribution (OOD) issues. For instance, masking high-magnitude frequencies or mixing a high-magnitude frequency with a low-magnitude one can drastically alter the original signal. We analyzed the features of augmented and original samples to highlight the OOD problem. Fig. 2 presents the t-SNE Van der Maaten & Hinton (2008) visualization of features of different augmented (orange) samples and original (blue) samples. Obvious domain gap can be found in full-spectrum perturbations such as FreqMask, FreqMix, and FreqAdd. Fig. 4 (c) illustrates the results of swapping (mixing) a minor frequency and a dominant frequency, which generates samples that are significantly away from the original data.

To tackle the OOD issue and align the distributions between augmented and original samples, we introduced Dominant Shuffle, which reduces perturbations during data augmentation. Let $F(\omega) = \text{FFT}([x, y])$ be the the complex frequency-domain representation of original time series, we perturb in the frequency domain by shuffling dominant frequency components. Let $\Omega_k = [\omega_1, \omega_2, ..., \omega_k]$ be the set of top-$k$ frequencies with highest magnitudes, then the dominant shuffle can be formulated as

$$\hat{F}(\omega) = \begin{cases} F(\hat{\omega}), \hat{\omega} \leftarrow \Omega_k, \text{ if } \omega \in \Omega_k \\ F(\omega) \qquad\qquad\quad otherwise, \end{cases} \tag{2}$$

where $\hat{\omega} \leftarrow \Omega$ denotes $\hat{\omega}$ is randomly chosen from $\Omega_k$ without replacement. $\hat{F}(\omega)$ is dominant shuffled frequency-domain representation, which is then converted back to time domain. Fig. 3 illustrates the process of shuffling $k = 3$ dominant frequencies.

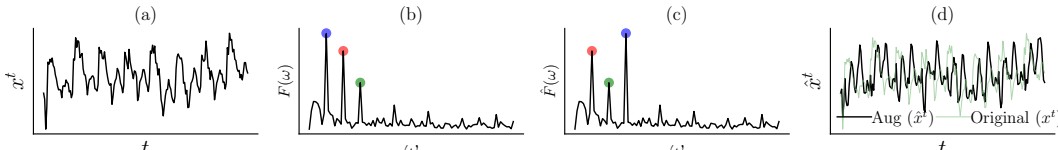

Figure 3: Illustration of shuffling three dominant frequencies. (a) The original time-series $x^t$. (b) and (c) Frequency-domain representations before and after dominant shuffle. Color dots represent the shuffle of dominant frequencies. (d) Augmented time series with original time series as reference.

We shuffle dominant frequencies because, as shown in Fig. 4 (a), this approach introduces moderate perturbations that result in valid augmentations. Perturbing minor frequencies, as demonstrated in Fig. 4 (b), has minimal impact on the original data and fails to produce effective augmentation. In contrast, as shown in Fig. 4 (c), swapping a dominant frequency with a minor one can significantly distort the original data, leading to out-of-distribution (OOD) issues. In conclusion, dominant shuffle applies moderate perturbations to the original data, generating effective augmented samples that improve the model's performance and robustness.

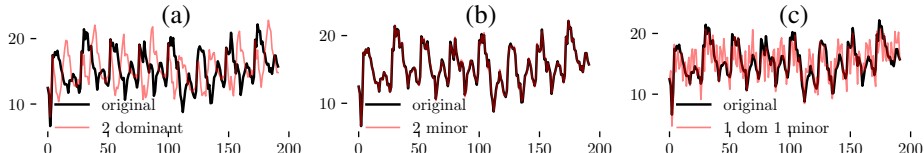

Figure 4: (a): swapping two dominant frequencies provide valid augmentations without excessive perturbations. (b): swapping minor frequencies barely changes the original data. (c): swapping a dominant frequency with a minor frequency introduces excessive perturbations.

# 4 EXPERIMENTS

In this section, we first introduce the implementation details in Sec. 4.1, and then compared the performance of various SOTA models with and without dominant shuffle in Sec. 4.2. In Sec. 4.3, we thoroughly compared dominant shuffle with various data augmentation methods. Finally, we conducted ablation studies to verify hyperparameter sensitivity and justify design choices in Sec. 4.4.

## 4.1 EXPERIMENTAL SETUPS

**Implementation details**   All the experiments were conducted with the PyTorch Paszke et al. (2019) framework on a single NVIDIA RTX 3090 GPU. We reproduced the results of other data augmentations using their official code or following the original papers. Please refer to appendix A.2 for the details about our reimplementations. We only changed the data augmentation for fair comparisons. Following the practice of Chen et al. (2023a), we performed data augmentations to double the size of the original training dataset unless otherwise specified.

**Evaluation protocols**   We tested our method with short-term and long-term prediction protocols. In the long-term protocol, the prediction period $T$ ranges from 96 to 720, with variations at 96, 192, 336, and 720. In contrast, the short-term protocol has prediction periods ranging from 12 to 48, with variations at 12, 24, 36, and 48. Following the common practice of previous works Zhou et al. (2021); Wu et al. (2021); Zhou et al. (2022b); Liu et al. (2024); Wang et al. (2023); Wu et al. (2023), we quantified the performance of the prediction using the mean-squared error (MSE) between the ground-truth and the prediction.

**Datasets**   For long-term prediction, we experimented on eight well-established benchmarks: the ETT datasets (ETTh1, ETTh2, ETTm1, ETTm2) Zhou et al. (2021), and the Weather, Electricity, Exchange, and Traffic datasets Wu et al. (2021). For short-term prediction, following iTransformer Liu et al. (2024), we used four public traffic network datasets (PEMS03, PEMS04, PEMS07, PEMS08) from PEMS Chen et al. (2001). Each dataset is divided into training, testing, and evaluation subsets in specific ratios. The training, testing, and evaluation ratio is 6:2:2 for ETT and PEMS datasets, and the ratio is 7:1:2 for Electricity, Traffic, Weather, and Exchange-rate datasets. Detailed statistics of these datasets are summarized in appendix A.1. For each setting (dataset+prediction length $T$), we tuned the optimal number of dominant frequencies $k$ on the evaluation set. The optimal $k$ on various datasets can be found in appendix B.4.

**Baseline Models**   We selected diverse models as the baseline in our experiments, including , Autoformer Wu et al. (2021), Lightts Zhang et al. (2022a), SCINet Liu et al. (2022), TiDE Das et al. (2023), MICN Wang et al. (2023), PatchTST Nie et al. (2023b), iTransformer Liu et al. (2024), PDF Dai et al. (2024), PathFormer Chen et al. (2024). For short-term prediction, we used the SOTA iTransformer Liu et al. (2024) on PEMS dataset Chen et al. (2001) as the baseline model.

**Other data augmentation methods**   We compared the proposed method with nine existing data augmentation methods, including three time-domain augmentations (ASD Forestier et al. (2017), MSB Bandara et al. (2021) Upsample  Semenoglou et al. (2023)), five frequency-domain methods (FreqMix Chen et al. (2023a), FreqMask Chen et al. (2023a), FreqAdd Zhang et al. (2022b), FreqPool Chen et al. (2023b), Robusttad Gao et al. (2020)), and a temporal-frequency method STAug Zhang et al. (2023).

## 4.2 COMPARISON WITH STATE-OF-THE-ARTS

We first evaluated the performance of several state-of-the-art time series prediction models with and without dominant shuffle. The averaged mean squared errors (MSE) across various prediction lengths (96, 192, 336, 720) is calculated for each dataset. The results in Fig. 5 clearly demonstrate that our

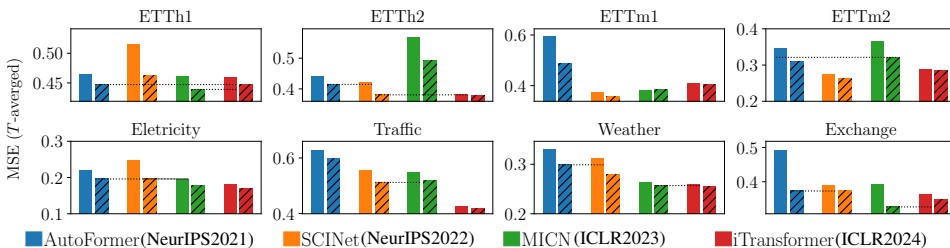

Figure 5: Performance of different models with (right striped bars) and without (left color bars) dominant shuffle. The horizontal dotted lines demonstrate how dominant shuffle helps one model outperforms a more advanced model.

method consistently reduces the prediction error for all the cases. In some cases, dominant shuffle surpasses even a highly sophisticated model. For example, on the ETTh1 dataset, our approach significantly improves the performance of AutoFormer Wu et al. (2021) and MICN Wang et al. (2023), and helps them outperform the latest iTransformer Liu et al. (2024) model. On the Exchange and Weather dataset, our approach enables AutoFormer to outperform SCINet Liu et al. (2022) and assists MICN Wang et al. (2023) in surpassing iTransformer Liu et al. (2024).

## 4.3 COMPARISONS WITH OTHER DATA AUGMENTATIONS

We compared different data augmentation methods on various datasets and baseline models under short-term, long-term, and few-shot time-series prediction settings. The performance are quantified by mean squared errors (MSE). Fig. 1 demonstrates the relative improvements (%) of various augmentation methods over the baseline. Tab. 1 and 2 summarize the performance of six representative models (MICN, SCINet, TiDE, and LightTS) on long-term prediction. Limited by the space, we only reported the MSE of six subsets (ETTh1, ETTh2, ETTm1, Electricity, Weather, and Exchange rate). The full results on all the eight subsets, together with the visualizations of some example predictions, and additional long-term prediction results with PatchTST Nie et al. (2023b), PatchFormer Chen et al. (2024), and PDF Dai et al. (2024) can be found in appendix B. The results of short-term prediction can be found in Tab. 3. We also evaluated our method with cold-start Chen et al. (2023a) and few-shot prediction Jin et al. (2024) settings where only very small proportion of training samples are available. The results can be found in appendix C. All the results are measured by the average performance of 5 runs with different random seeds, and some of the standard deviations can be found in appendix B.5. In all the tables, the best results of each setting are highlighted in bold text. We merged the results of FreqMix and FreqMask by selecting the superior one in each case. The merged results are denoted as 'MixMask'.

As demonstrated in Tab. 1 and 2, our method consistently improves the baseline on 96% of the cases, while other augmentation methods, e.g. FreqMix, outperform the baseline for around 87% of the cases. Our method also outperforms other augmentation methods on more than 77% of the cases. Moreover, our method achieves larger relative improvements as the prediction length $T$ increased, highlighting its strong capacity in long-term predictions. Tab. 3 summarizes the MSE of short-term prediction using the iTransformer Liu et al. (2024) model on the PEMS datasets Chen et al. (2001). The prediction errors are generally lower than the errors in long-term prediction. Our method outperforms other augmentations in most cases, although the improvements are marginal compared to long-term prediction. This is because short-term prediction is relatively easy, and the performance has already reached a saturation. The fewshot and cold-start results in appendix C demonstrate demonstrate the strong generalizability of our method with limited training samples.

| Method | ETTh1 | | | | ETTh2 | | | | ETTm1 | | | |
|---|---|---|---|---|---|---|---|---|---|---|---|---|
| | 96 | 192 | 336 | 720 | 96 | 192 | 336 | 720 | 96 | 192 | 336 | 720 |
| **iTransformer** | | | | | | | | | | | | |
| Baseline | 0.392 | 0.447 | 0.483 | 0.516 | 0.303 | 0.381 | 0.412 | 0.434 | 0.344 | 0.383 | 0.421 | 0.494 |
| ASD | 0.398 | 0.456 | 0.483 | 0.512 | 0.310 | 0.388 | 0.432 | 0.452 | 0.340 | 0.382 | 0.454 | 0.492 |
| MSB | 0.387 | 0.460 | 0.494 | 0.531 | 0.309 | 0.382 | 0.447 | 0.433 | 0.339 | 0.386 | 0.467 | 0.510 |
| Upsample | 0.391 | 0.445 | 0.481 | 0.519 | 0.305 | 0.381 | 0.419 | 0.430 | 0.351 | 0.381 | 0.432 | 0.489 |
| FreqAdd | 0.389 | 0.446 | 0.475 | 0.510 | 0.300 | 0.384 | 0.416 | 0.438 | 0.350 | 0.385 | 0.422 | 0.490 |
| FreqPool | 0.433 | 0.456 | 0.497 | 0.532 | 0.313 | 0.392 | 0.415 | 0.450 | 0.347 | 0.392 | 0.430 | 0.499 |
| Robusttad | 0.390 | 0.445 | 0.497 | 0.510 | 0.312 | 0.388 | 0.412 | 0.439 | 0.353 | 0.382 | 0.421 | 0.498 |
| STAug | 0.390 | 0.445 | 0.489 | 0.511 | 0.323 | 0.428 | 0.486 | 0.483 | 0.339 | 0.383 | **0.417** | 0.485 |
| MixMask | 0.388 | 0.440 | 0.477 | 0.504 | 0.301 | **0.380** | 0.414 | 0.434 | 0.334 | 0.375 | 0.421 | **0.485** |
| Ours | **0.383** | **0.438** | **0.473** | **0.492** | **0.298** | 0.382 | **0.411** | **0.428** | **0.332** | **0.374** | 0.424 | 0.492 |
| **AutoFormer** | | | | | | | | | | | | |
| Baseline | 0.429 | 0.440 | 0.495 | 0.498 | 0.381 | 0.443 | 0.471 | 0.475 | 0.467 | 0.610 | 0.529 | 0.773 |
| ASD | 0.450 | 0.485 | 0.523 | 0.556 | 0.370 | 0.465 | 0.476 | 0.503 | 0.480 | 0.620 | 0.502 | 0.633 |
| MSB | 0.462 | 0.517 | 0.612 | 0.579 | 0.434 | 0.523 | 0.556 | 0.462 | 0.499 | 0.645 | 0.553 | 0.721 |
| Upsample | 0.416 | 0.523 | 0.480 | **0.482** | 0.353 | 0.460 | 0.455 | 0.509 | 0.498 | 0.630 | 0.512 | 0.667 |
| FreqAdd | 0.460 | 0.487 | 0.497 | 0.525 | 0.367 | 0.439 | 0.480 | 0.504 | 0.419 | 0.554 | 0.546 | 0.569 |
| FreqPool | 0.446 | 0.457 | 0.523 | 0.512 | 0.392 | 0.442 | 0.470 | 0.493 | 0.479 | 0.623 | 0.510 | 0.754 |
| Robusttad | 0.437 | 0.452 | 0.492 | 0.477 | 0.367 | 0.497 | 0.502 | 0.527 | 0.432 | 0.510 | 0.553 | 0.623 |
| STAug | 0.429 | 0.478 | 0.505 | 0.506 | 0.354 | 0.443 | 0.496 | 0.495 | 0.415 | 0.581 | 0.588 | 0.693 |
| MixMask | 0.420 | 0.445 | 0.467 | 0.474 | 0.358 | 0.421 | 0.470 | 0.467 | 0.415 | 0.510 | **0.491** | 0.588 |
| Ours | **0.409** | **0.436** | **0.458** | 0.486 | **0.335** | **0.419** | **0.453** | **0.452** | **0.392** | **0.506** | **0.491** | **0.559** |
| **MICN** | | | | | | | | | | | | |
| Baseline | 0.384 | 0.425 | 0.464 | 0.574 | 0.358 | 0.518 | 0.566 | 0.827 | 0.313 | 0.360 | 0.389 | 0.461 |
| ASD | 0.380 | 0.430 | 0.472 | 0.523 | 0.377 | 0.539 | 0.620 | 0.843 | 0.315 | 0.362 | 0.399 | 0.457 |
| MSB | 0.423 | 0.423 | 0.501 | 0.559 | 0.402 | 0.623 | 0.790 | 1.126 | 0.330 | 0.358 | 0.402 | 0.459 |
| Upsample | 0.396 | 0.435 | 0.463 | 0.550 | 0.366 | 0.500 | 0.831 | 0.752 | 0.339 | 0.377 | 0.402 | 0.475 |
| FreqAdd | 0.390 | 0.430 | 0.477 | 0.643 | 0.370 | 0.521 | 0.626 | 0.975 | 0.316 | 0.360 | 0.407 | 0.478 |
| FreqPool | 0.399 | 0.465 | 0.473 | 0.572 | 0.365 | 0.553 | 0.550 | 0.812 | 0.336 | 0.372 | 0.397 | 0.466 |
| Robusttad | 0.392 | 0.436 | 0.491 | 0.556 | 0.339 | 0.529 | 0.553 | 0.998 | 0.339 | 0.359 | 0.396 | 0.472 |
| STAug | 0.374 | 0.429 | 0.489 | 0.608 | 0.413 | 0.760 | 1.330 | 2.608 | 0.313 | 0.360 | 0.418 | 0.483 |
| MixMask | 0.378 | 0.423 | 0.461 | 0.521 | 0.339 | 0.488 | 0.544 | 0.735 | **0.301** | 0.352 | 0.401 | **0.454** |
| Ours | **0.373** | **0.421** | **0.452** | **0.510** | **0.310** | **0.427** | **0.507** | **0.731** | 0.314 | 0.360 | **0.387** | 0.470 |
| **SCINet** | | | | | | | | | | | | |
| Baseline | 0.485 | 0.506 | 0.519 | 0.552 | 0.372 | 0.416 | 0.429 | 0.470 | 0.316 | 0.353 | 0.387 | 0.431 |
| ASD | 0.494 | 0.480 | 0.491 | 0.559 | 0.362 | 0.402 | 0.432 | 0.499 | 0.331 | 0.367 | 0.389 | 0.453 |
| MSB | 0.489 | 0.466 | 0.502 | 0.547 | 0.359 | 0.396 | 0.458 | 0.476 | 0.320 | 0.351 | 0.396 | 0.478 |
| Upsample | 0.471 | 0.457 | 0.479 | 0.541 | 0.379 | 0.407 | 0.403 | 0.482 | 0.342 | 0.386 | 0.399 | 0.442 |
| FreqAdd | 0.428 | 0.452 | 0.469 | 0.532 | 0.335 | 0.385 | 0.403 | 0.447 | 0.304 | **0.338** | 0.373 | 0.421 |
| FreqPool | 0.499 | 0.510 | 0.557 | 0.549 | 0.410 | 0.453 | 0.432 | 0.475 | 0.331 | 0.362 | 0.379 | 0.432 |
| Robusttad | 0.462 | 0.501 | 0.498 | 0.559 | 0.362 | 0.431 | 0.419 | 0.496 | 0.331 | 0.351 | 0.394 | 0.438 |
| STAug | 0.457 | 0.500 | 0.524 | 0.534 | 0.538 | 0.636 | 0.681 | 0.648 | 0.319 | 0.357 | 0.389 | 0.445 |
| MixMask | 0.427 | 0.452 | 0.465 | 0.548 | **0.335** | 0.377 | 0.400 | 0.438 | **0.302** | 0.341 | 0.376 | 0.423 |
| Ours | **0.417** | **0.443** | **0.461** | **0.527** | **0.335** | **0.375** | **0.392** | **0.421** | **0.302** | **0.338** | **0.372** | **0.420** |
| **TiDE** | | | | | | | | | | | | |
| Baseline | 0.401 | 0.434 | 0.521 | 0.558 | 0.304 | 0.350 | 0.331 | 0.399 | 0.311 | 0.340 | 0.366 | 0.420 |
| ASD | 0.417 | 0.441 | 0.513 | 0.556 | 0.320 | 0.351 | 0.367 | 0.422 | 0.319 | 0.341 | 0.399 | 0.432 |
| MSB | 0.422 | 0.476 | 0.529 | 0.579 | 0.331 | 0.379 | 0.334 | 0.401 | 0.302 | 0.356 | 0.382 | 0.451 |
| Upsample | 0.431 | 0.452 | 0.533 | 0.604 | 0.346 | 0.372 | 0.350 | 0.456 | 0.324 | 0.339 | 0.378 | 0.463 |
| FreqAdd | 0.385 | 0.420 | 0.477 | 0.505 | 0.289 | 0.336 | 0.330 | 0.390 | 0.309 | 0.339 | 0.365 | 0.417 |
| FreqPool | 0.423 | 0.455 | 0.510 | 0.592 | 0.312 | .376 | 0.339 | 0.397 | 0.319 | 0.352 | 0.397 | 0.453 |
| Robusttad | 0.396 | 0.432 | 0.521 | 0.537 | 0.331 | 0.352 | 0.337 | 0.398 | 0.321 | 0.346 | 0.382 | 0.437 |
| STAug | 0.515 | 0.535 | 0.521 | 0.558 | 0.390 | 0.437 | 0.403 | 0.508 | 0.310 | 0.337 | **0.364** | 0.417 |
| MixMask | **0.385** | 0.420 | 0.478 | 0.507 | 0.289 | 0.339 | 0.330 | 0.391 | 0.299 | 0.332 | 0.367 | 0.416 |
| Ours | **0.385** | **0.414** | **0.467** | **0.498** | **0.283** | **0.332** | **0.324** | **0.388** | **0.297** | **0.328** | 0.365 | **0.412** |
| **LightTS** | | | | | | | | | | | | |
| Baseline | 0.448 | 0.444 | 0.663 | 0.706 | 0.369 | 0.476 | 0.738 | 1.165 | 0.323 | 0.347 | 0.428 | 0.476 |
| ASD | 0.451 | 0.476 | 0.633 | 0.681 | 0.392 | 0.469 | 0.701 | 0.998 | 0.356 | 0.352 | 0.441 | 0.478 |
| MSB | 0.467 | 0.463 | 0.627 | 0.652 | 0.378 | 0.472 | 0.652 | 1.123 | 0.371 | 0.349 | 0.430 | 0.479 |
| Upsample | 0.449 | 0.472 | 0.610 | 0.637 | 0.401 | 0.487 | 0.714 | 1.245 | 0.329 | 0.366 | 0.453 | 0.492 |
| FreqAdd | 0.417 | 0.430 | 0.578 | 0.622 | 0.351 | 0.453 | 0.689 | 1.125 | 0.322 | 0.352 | 0.400 | 0.450 |
| FreqPool | 0.463 | 0.471 | 0.652 | 0.690 | 0.369 | 0.512 | 0.723 | 1.264 | 0.336 | 0.351 | 0.442 | 0.497 |
| Robusttad | 0.445 | 0.442 | 0.590 | 0.654 | 0.372 | 0.468 | 0.699 | 0.982 | 0.331 | 0.352 | 0.441 | 0.462 |
| STAug | 0.445 | 0.441 | 0.669 | 0.714 | 0.520 | 0.807 | 2.101 | 2.467 | 0.320 | 0.343 | 0.427 | 0.476 |
| MixMask | 0.417 | 0.429 | 0.575 | 0.620 | 0.337 | 0.426 | 0.643 | 0.993 | **0.316** | **0.340** | 0.398 | 0.447 |
| Ours | **0.405** | **0.423** | **0.565** | **0.603** | **0.335** | **0.395** | **0.575** | **0.827** | 0.322 | **0.340** | **0.391** | **0.440** |

Table 1: MSE of the long-term prediction on the ETT Zhou et al. (2021) datasets. The best values are in bold text.

## 4.4 ABLATION STUDY

Our method includes a hyper-parameter $k$ and two unique designs: 1) perturb the dominant frequencies and 2) shuffle the dominant frequency components. We conducted ablation studies to investigate the impact of hyperparameters and to justify our design choices.

| Method | | Electricity | | | | Weather | | | | Exchange Rate | | | |
|---|---|---|---|---|---|---|---|---|---|---|---|---|---|
| | | 96 | 192 | 336 | 720 | 96 | 192 | 336 | 720 | 96 | 192 | 336 | 720 |
| iTransformer | Baseline | 0.152 | 0.159 | 0.179 | 0.230 | 0.175 | 0.224 | 0.281 | 0.362 | **0.086** | 0.180 | 0.335 | 0.856 |
| | ASD | 0.173 | 0.179 | 0.201 | 0.234 | 0.191 | 0.223 | 0.280 | 0.364 | 0.088 | 0.183 | 0.343 | 0.872 |
| | MSB | 0.182 | 0.182 | 0.194 | 0.267 | 0.185 | 0.235 | 0.284 | 0.359 | 0.089 | 0.189 | 0.359 | 0.907 |
| | Upsample | 0.166 | 0.188 | 0.216 | 0.221 | 0.204 | 0.257 | 0.291 | 0.373 | 0.086 | 0.180 | 0.338 | 0.834 |
| | FreqAdd | **0.150** | 0.157 | 0.172 | 0.204 | 0.181 | 0.230 | 0.285 | 0.362 | 0.087 | 0.181 | 0.333 | 0.837 |
| | FreqPool | 0.169 | 0.170 | 0.194 | 0.237 | 0.184 | 0.223 | 0.279 | 0.378 | 0.088 | 0.183 | 0.330 | 0.832 |
| | Robusttad | 0.150 | 0.157 | 0.176 | 0.210 | 0.172 | 0.225 | 0.281 | 0.357 | 0.087 | 0.179 | 0.329 | 0.833 |
| | STAug | 0.160 | 0.173 | 0.218 | 0.372 | 0.206 | 0.264 | 0.319 | 0.385 | 0.086 | 0.178 | 0.335 | 0.866 |
| | MixMask | 0.151 | 0.158 | 0.173 | 0.205 | 0.175 | 0.224 | 0.279 | 0.354 | 0.089 | 0.178 | 0.328 | 0.845 |
| | Ours | **0.150** | **0.156** | **0.171** | **0.199** | **0.171** | **0.221** | **0.276** | **0.351** | **0.086** | **0.176** | **0.313** | **0.821** |
| AutoFormer | Baseline | 0.203 | 0.208 | 0.231 | 0.239 | 0.241 | 0.314 | 0.341 | 0.425 | 0.143 | 0.305 | 0.470 | 1.056 |
| | ASD | 0.247 | 0.216 | 0.221 | 0.235 | 0.652 | 0.392 | 0.416 | 0.513 | 0.141 | 0.280 | 0.579 | 1.240 |
| | MSB | 0.237 | 0.256 | 0.295 | 0.236 | 0.256 | 0.379 | 0.402 | 0.468 | 0.156 | 0.254 | 0.513 | 1.339 |
| | Upsample | 0.201 | 0.209 | 0.232 | 0.268 | 0.281 | 0.294 | 0.329 | 0.385 | 0.141 | 0.292 | 0.553 | 1.295 |
| | FreqAdd | 0.193 | 0.197 | 0.212 | 0.225 | 0.255 | 0.323 | 0.370 | 0.419 | 0.143 | 0.369 | 0.716 | 1.173 |
| | FreqPool | 0.213 | 0.224 | 0.234 | 0.257 | 0.237 | 0.339 | 0.372 | 0.446 | 0.142 | 0.336 | 0.532 | 1.014 |
| | Robusttad | 0.230 | 0.242 | 0.261 | 0.231 | 0.27 | 0.334 | 0.351 | 0.429 | 0.142 | 0.309 | 0.462 | 1.123 |
| | STAug | 0.191 | 0.206 | 0.217 | 0.234 | 0.250 | 0.300 | 0.347 | 0.418 | 0.140 | 0.326 | 0.594 | 1.176 |
| | MixMask | 0.177 | 0.194 | 0.206 | 0.224 | 0.240 | 0.302 | 0.330 | 0.422 | 0.141 | 0.284 | 0.453 | 0.778 |
| | Ours | **0.171** | **0.191** | **0.203** | **0.219** | **0.214** | **0.273** | **0.327** | **0.383** | **0.136** | **0.243** | **0.418** | **0.695** |
| MICN | Baseline | 0.171 | 0.183 | 0.198 | 0.224 | 0.188 | 0.241 | 0.278 | 0.350 | 0.091 | 0.185 | 0.355 | 0.941 |
| | ASD | 0.165 | 0.174 | 0.190 | 0.237 | 0.189 | 0.242 | 0.276 | 0.354 | 0.087 | 0.175 | 0.337 | 1.203 |
| | MSB | 0.179 | 0.182 | 0.201 | 0.225 | 0.201 | 0.250 | 0.291 | 0.365 | 0.088 | 0.176 | 0.360 | 0.995 |
| | Upsample | 0.182 | 0.180 | 0.203 | 0.220 | 0.193 | 0.249 | 0.279 | 0.372 | 0.084 | 0.171 | 0.313 | **0.702** |
| | FreqAdd | 0.160 | 0.169 | 0.182 | 0.199 | 0.180 | 0.234 | 0.282 | 0.350 | 0.087 | 0.174 | 0.349 | 0.923 |
| | FreqPool | 0.182 | 0.203 | 0.241 | 0.256 | 0.192 | 0.257 | 0.278 | 0.351 | 0.089 | 0.179 | 0.394 | 0.923 |
| | Robusttad | 0.179 | 0.220 | 0.234 | 0.227 | 0.192 | 0.239 | 0.292 | 0.343 | 0.085 | 0.179 | 0.336 | 0.932 |
| | STAug | 0.180 | 0.195 | 0.210 | 0.224 | 0.272 | 0.356 | 0.433 | 0.559 | 0.092 | 0.183 | 0.313 | 0.790 |
| | MixMask | 0.159 | **0.165** | **0.178** | **0.195** | 0.185 | 0.239 | 0.281 | 0.344 | 0.086 | 0.174 | 0.337 | 0.796 |
| | Ours | **0.157** | 0.168 | 0.178 | 0.211 | **0.179** | **0.232** | **0.275** | **0.342** | **0.084** | **0.169** | **0.303** | 0.750 |
| SCINet | Baseline | 0.212 | 0.237 | 0.255 | 0.286 | 0.229 | 0.282 | 0.334 | 0.402 | 0.099 | 0.191 | 0.356 | 0.916 |
| | ASD | 0.229 | 0.241 | 0.239 | 0.282 | 0.254 | 0.276 | 0.356 | 0.462 | 0.095 | 0.204 | 0.379 | 1.230 |
| | MSB | 0.232 | 0.237 | 0.228 | 0.274 | 0.279 | 0.265 | 0.374 | 0.454 | 0.093 | 0.267 | 0.402 | 0.965 |
| | Upsample | 0.250 | 0.232 | 0.271 | 0.309 | 0.243 | 0.299 | 0.361 | 0.431 | 0.092 | 0.196 | **0.311** | 0.932 |
| | FreqAdd | 0.176 | 0.195 | 0.212 | 0.237 | 0.208 | 0.258 | 0.309 | 0.385 | 0.092 | 0.186 | 0.343 | 0.920 |
| | FreqPool | 0.230 | 0.221 | 0.242 | 0.339 | 0.261 | 0.290 | 0.337 | 0.456 | 0.096 | 0.183 | 0.551 | 0.938 |
| | Robusttad | 0.189 | 0.202 | 0.210 | 0.243 | 0.229 | 0.281 | 0.331 | 0.410 | 0.093 | 0.186 | 0.334 | 0.957 |
| | STAug | 0.210 | 0.239 | 0.282 | 0.411 | 0.277 | 0.329 | 0.372 | 0.435 | 0.098 | 0.191 | 0.342 | 0.931 |
| | MixMask | **0.171** | **0.188** | 0.204 | 0.230 | 0.205 | 0.250 | 0.310 | **0.374** | 0.093 | 0.179 | 0.336 | 0.928 |
| | Ours | 0.172 | **0.188** | **0.200** | **0.225** | **0.197** | **0.246** | **0.299** | 0.379 | **0.091** | **0.175** | 0.342 | **0.890** |
| TiDE | Baseline | 0.207 | 0.197 | 0.211 | 0.238 | 0.177 | 0.220 | 0.265 | 0.323 | 0.093 | 0.184 | 0.330 | 0.860 |
| | ASD | 0.232 | 0.220 | 0.231 | 0.265 | 0.189 | 0.221 | 0.297 | 0.332 | 0.095 | 0.206 | 0.351 | 0.962 |
| | MSB | 0.210 | 0.219 | 0.253 | 0.261 | 0.199 | 0.254 | 0.273 | 0.339 | 0.092 | 0.179 | 0.358 | 0.941 |
| | Upsample | 0.206 | 0.199 | 0.223 | 0.274 | 0.203 | 0.267 | 0.331 | 0.355 | 0.091 | 0.182 | 0.331 | 0.852 |
| | FreqAdd | 0.150 | 0.163 | 0.177 | 0.209 | **0.173** | **0.216** | 0.263 | **0.322** | **0.088** | 0.180 | 0.330 | 0.848 |
| | FreqPool | 0.224 | 0.238 | 0.233 | 0.270 | 0.189 | 0.224 | 0.292 | 0.334 | 0.092 | 0.334 | 0.521 | 1.124 |
| | Robusttad | 0.176 | 0.166 | 0.182 | 0.229 | 0.182 | 0.231 | 0.279 | 0.330 | 0.099 | 0.232 | 0.331 | 0.924 |
| | STAug | 0.230 | 0.210 | 0.192 | 0.225 | 0.205 | 0.247 | 0.292 | 0.364 | 0.092 | 0.184 | 0.330 | 0.859 |
| | MixMask | **0.143** | 0.155 | **0.164** | 0.210 | **0.173** | **0.216** | 0.263 | 0.323 | 0.089 | 0.180 | 0.329 | 0.861 |
| | Ours | **0.143** | **0.150** | 0.165 | **0.202** | 0.177 | 0.219 | **0.261** | **0.322** | **0.088** | 0.179 | 0.324 | **0.847** |
| LightTS | Baseline | 0.210 | 0.169 | 0.182 | 0.212 | 0.168 | 0.210 | 0.260 | 0.320 | 0.139 | 0.252 | 0.412 | 0.840 |
| | ASD | 0.225 | 0.179 | 0.198 | 0.232 | 0.179 | 0.210 | 0.271 | 0.321 | 0.132 | 0.320 | 0.436 | 1.036 |
| | MSB | 0.233 | 0.182 | 0.204 | 0.228 | 0.170 | 0.214 | 0.259 | 0.332 | 0.117 | 0.294 | 0.502 | 0.964 |
| | Upsample | 0.246 | 0.179 | 0.211 | 0.254 | 0.182 | 0.223 | 0.257 | 0.336 | 0.099 | 0.251 | 0.369 | 0.702 |
| | FreqAdd | 0.213 | 0.159 | 0.177 | 0.210 | 0.164 | 0.207 | 0.258 | 0.317 | 0.098 | 0.522 | 0.565 | 1.583 |
| | FreqPool | 0.219 | 0.174 | 0.197 | 0.236 | 0.193 | 0.254 | 0.267 | 0.339 | 0.099 | 0.275 | 0.394 | 0.793 |
| | Robusttad | 0.212 | 0.169 | 0.181 | 0.223 | 0.172 | 0.223 | 0.259 | 0.324 | 0.092 | 0.279 | 0.451 | 0.796 |
| | STAug | 0.224 | 0.267 | 0.294 | 0.351 | 0.214 | 0.263 | 0.382 | 0.371 | 0.096 | **0.212** | 0.380 | 0.690 |
| | MixMask | **0.192** | 0.158 | 0.175 | 0.211 | **0.163** | 0.206 | 0.257 | 0.318 | 0.099 | 0.384 | 0.518 | 0.774 |
| | Ours | 0.210 | **0.156** | **0.173** | **0.206** | 0.165 | **0.205** | **0.249** | **0.312** | **0.088** | 0.243 | **0.361** | **0.676** |

Table 2: MSE of the long-term prediction on the Weather, Electricity, and Exchange Rate Wu et al. (2021) datasets. The best values are marked with bold text.

### 4.4.1 NUMBER OF DOMINANT FREQUENCIES

The only hyper-parameter in our method is the number of dominant frequencies ($k$) to be shuffled. We evaluated the performance using various $k$ values with iTransformer Liu et al. (2024). The results in Fig. 6 reveal that larger $k$ generally leads to worse results, this is because larger $k$ increases the

| Methods | PEMS03 | | | | PEMS04 | | | | PEMS07 | | | |
|---|---|---|---|---|---|---|---|---|---|---|---|---|
| | 12 | 24 | 36 | 48 | 12 | 24 | 36 | 48 | 12 | 24 | 36 | 48 |
| Baseline | 0.070 | 0.097 | 0.134 | 0.164 | 0.088 | 0.124 | 0.160 | 0.196 | 0.067 | 0.097 | 0.128 | 0.156 |
| ASD | 0.072 | 0.096 | 0.152 | 0.239 | 0.098 | 0.132 | 0.156 | 0.190 | 0.069 | 0.099 | 0.154 | 0.181 |
| MSB | 0.096 | 0.131 | 0.129 | 0.214 | 0.087 | 0.134 | 0.167 | 0.219 | 0.098 | 0.096 | 0.137 | 0.165 |
| Upsample | 0.069 | 0.096 | 0.128 | 0.179 | 0.087 | 0.124 | 0.158 | 0.199 | 0.072 | 0.099 | 0.127 | 0.155 |
| FreqAdd | 1.036 | 0.104 | 0.251 | 0.362 | 0.088 | 0.125 | 0.159 | 0.201 | 0.067 | 0.097 | 0.127 | 0.155 |
| FreqPool | 1.234 | 0.178 | 0.296 | 0.451 | 0.099 | 0.145 | 0.178 | 0.226 | 0.079 | 0.104 | 0.152 | 0.172 |
| Robusttad | 0.082 | 0.098 | 0.132 | 1.520 | 0.089 | 0.123 | 0.161 | 0.195 | 0.067 | 0.097 | 0.129 | 0.157 |
| STAug | 0.079 | 0.112 | 0.195 | 0.456 | 0.087 | 0.120 | 0.162 | 0.304 | 0.066 | 0.096 | 0.132 | 0.165 |
| Mask | 0.443 | 1.205 | 0.233 | 1.510 | 0.086 | 0.119 | 0.158 | 0.346 | **0.065** | 0.095 | 0.125 | 0.156 |
| Mix | 1.018 | 0.097 | 0.877 | 1.501 | **0.085** | 0.119 | 0.154 | 0.205 | **0.065** | **0.094** | 0.134 | 0.152 |
| Ours | **0.067** | **0.095** | **0.126** | 0.235 | **0.085** | **0.118** | **0.149** | **0.182** | **0.065** | **0.094** | **0.123** | **0.148** |

Table 3: Short-term prediction using the iTransformer Liu et al. (2024) on the PEMS datasets Chen et al. (2001).

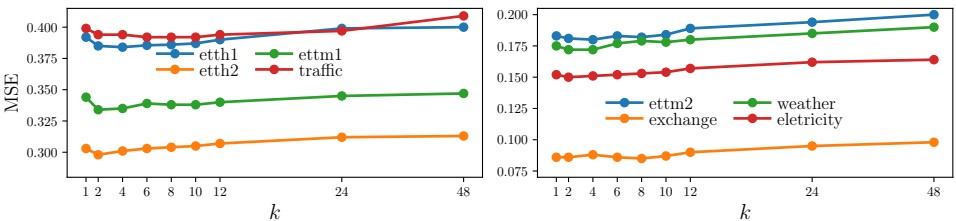

Figure 6: MSE of various $k$ values on four datasets under the predict-96 setting. Our method is stable against $k$, and the performance varies slightly.

possibility of swapping a high-magnitude dominant frequency with a low-magnitude minor frequency, which introduces excessive perturbations and generates OOD samples.

### 4.4.2 SHUFFLE THE DOMINANT FREQUENCIES

In this experiment, we compared the combination of different perturbation strategies and operations.

We first compared perturbing different frequency proportions including dominant frequencies, minor frequencies, and the full spectrum. The results in Tab. 4 clearly indicate that perturbing the dominant frequencies significantly outperforms other options, while perturbing the minor frequencies yields the worst performance. Tab. 5 compares different perturbation operations including masking Chen et al. (2023a), adding noise Gao et al. (2020); Lim & Zohren (2021), randomization, and shuffling (ours). Shuffle consistently surpasses other operations in most of the cases.

| | | | ETTh1 | | | | ETTm2 | | | | Weather | | | |
|---|---|---|---|---|---|---|---|---|---|---|---|---|---|---|
| | | | 96 | 192 | 336 | 720 | 96 | 192 | 336 | 720 | 96 | 192 | 336 | 720 |
| iTrans | Shuffle | full | 0.391 | 0.447 | 0.486 | 0.509 | 0.182 | 0.247 | 0.311 | **0.403** | 0.175 | 0.223 | 0.278 | 0.355 |
| | | min | 0.389 | 0.445 | 0.494 | 0.505 | 0.181 | 0.251 | 0.310 | 0.413 | 0.174 | 0.225 | 0.282 | 0.355 |
| | | dom | **0.383** | **0.438** | **0.473** | **0.492** | **0.178** | 0.246 | 0.309 | 0.409 | **0.171** | 0.221 | **0.276** | **0.351** |
| | Mask | full | 0.390 | **0.442** | **0.475** | 0.503 | **0.179** | **0.251** | 0.311 | 0.411 | 0.178 | 0.228 | 0.284 | 0.359 |
| | | min | 0.389 | 0.444 | 0.487 | **0.499** | 0.183 | 0.252 | 0.311 | 0.412 | 0.180 | 0.226 | 0.282 | 0.361 |
| | | dom | 0.388 | 0.442 | 0.486 | 0.505 | 0.180 | 0.251 | 0.309 | 0.410 | 0.173 | 0.224 | 0.280 | 0.356 |
| MICN | Shuffle | full | 0.385 | 0.427 | 0.466 | 0.604 | 0.184 | 0.293 | 0.375 | 0.594 | 0.182 | 0.239 | 0.280 | 0.348 |
| | | min | 0.390 | 0.430 | 0.480 | 0.565 | 0.191 | 0.281 | 0.365 | 0.580 | 0.197 | 0.236 | 0.283 | 0.349 |
| | | dom | **0.373** | **0.421** | **0.452** | **0.510** | 0.174 | 0.263 | 0.348 | 0.502 | 0.179 | **0.232** | 0.275 | 0.342 |
| | Mask | full | 0.381 | 0.424 | 0.460 | 0.543 | 0.184 | 0.265 | 0.353 | 0.510 | 0.190 | 0.236 | **0.281** | 0.345 |
| | | min | 0.385 | 0.426 | 0.472 | 0.553 | 0.187 | 0.276 | 0.359 | 0.542 | 0.179 | 0.240 | **0.281** | 0.344 |
| | | dom | 0.377 | 0.421 | 0.454 | 0.543 | **0.175** | 0.268 | **0.337** | 0.505 | 0.178 | 0.239 | 0.283 | **0.342** |
| Lightts | Shuffle | full | 0.415 | 0.426 | 0.577 | 0.621 | 0.202 | **0.235** | 0.325 | 0.445 | **0.163** | **0.205** | 0.251 | 0.317 |
| | | min | 0.418 | 0.432 | 0.577 | 0.619 | 0.206 | 0.239 | 0.326 | 0.444 | 0.164 | 0.212 | 0.259 | 0.317 |
| | | dom | **0.405** | **0.423** | **0.565** | **0.603** | **0.195** | 0.245 | 0.312 | 0.422 | 0.165 | **0.205** | 0.249 | 0.312 |
| | Mask | full | **0.418** | 0.432 | **0.573** | 0.621 | 0.204 | **0.238** | 0.321 | 0.435 | 0.163 | 0.206 | 0.258 | **0.317** |
| | | min | 0.419 | 0.433 | 0.578 | 0.621 | 0.205 | 0.233 | 0.324 | 0.452 | 0.163 | 0.208 | 0.260 | **0.317** |
| | | dom | **0.418** | **0.424** | 0.579 | **0.618** | 0.198 | 0.240 | 0.312 | 0.430 | 0.162 | 0.201 | 0.250 | 0.317 |

Table 4: Comparison of perturbing different spectrum (full, minor, and dominant) using shuffle and random mask. Perturbing the dominant frequencies performs significantly better than perturbing other frequencies. And shuffle is also more effective than random mask.

| | | ETTh1 | | | | ETTm2 | | | | Weather | | | |
|---|---|---|---|---|---|---|---|---|---|---|---|---|---|
| | | 96 | 192 | 336 | 720 | 96 | 192 | 336 | 720 | 96 | 192 | 336 | 720 |
| iTrans | Mask | 0.388 | 0.442 | 0.486 | 0.505 | 0.180 | 0.251 | **0.309** | 0.410 | 0.173 | 0.224 | 0.280 | 0.356 |
| | Noise | 0.387 | 0.445 | 0.482 | 0.510 | 0.180 | 0.256 | 0.312 | 0.409 | 0.177 | 0.222 | 0.281 | 0.359 |
| | Random | 0.386 | 0.440 | 0.479 | 0.499 | 0.183 | 0.254 | 0.311 | **0.407** | **0.171** | 0.222 | 0.280 | 0.358 |
| | Shuffle | **0.383** | **0.438** | **0.473** | **0.492** | **0.178** | **0.246** | 0.309 | 0.409 | **0.171** | **0.221** | **0.276** | **0.351** |
| MICN | Mask | 0.377 | **0.421** | 0.454 | 0.543 | 0.175 | 0.268 | **0.337** | 0.505 | **0.178** | 0.239 | 0.283 | **0.342** |
| | Noise | 0.393 | 0.430 | 0.479 | 0.531 | 0.201 | 0.331 | 0.366 | 0.561 | 0.201 | 0.236 | 0.281 | 0.351 |
| | Random | 0.381 | 0.423 | 0.476 | 0.670 | 0.183 | 0.284 | 0.367 | 0.614 | 0.182 | 0.233 | 0.282 | 0.349 |
| | Shuffle | **0.373** | **0.421** | **0.452** | **0.510** | **0.174** | **0.263** | 0.348 | **0.502** | 0.179 | **0.232** | **0.275** | **0.342** |
| Lightts | Mask | 0.418 | 0.424 | 0.579 | 0.618 | 0.198 | 0.240 | **0.312** | 0.430 | **0.162** | **0.201** | 0.250 | 0.317 |
| | Noise | 0.432 | 0.451 | 0.566 | 0.636 | 0.221 | **0.236** | 0.351 | 0.433 | 0.169 | 0.219 | 0.259 | 0.321 |
| | Random | 0.414 | 0.431 | 0.570 | 0.610 | 0.206 | 0.244 | 0.324 | 0.442 | 0.171 | 0.213 | 0.263 | 0.323 |
| | Shuffle | **0.405** | **0.423** | **0.565** | **0.603** | **0.195** | 0.245 | **0.312** | **0.422** | 0.165 | 0.205 | **0.249** | **0.312** |

Table 5: Comparison of different dominant frequency perturbations. Shuffle outperforms other alternatives with clear margins.

The results in Tab. 4 and 5 justified the design decisions in *dominant shuffle* and confirm that both perturbing dominant frequencies and the shuffle operation is superior to other alternatives. More details about the experiments, including how we defined minor frequencies and we implemented mask, noise, and randomization perturbations can be found in appendix A.2.

### 4.4.3 DIFFERENT AUGMENTATION SIZES

In prior experiments, data augmentation was performed to doubled the original datasets. In this experiment, we assessed the performance of various augmentation sizes. The results with a larger augmentation size reflects the domain gap between augmented and original data, as larger augmentation sizes could introduce more OOD samples.

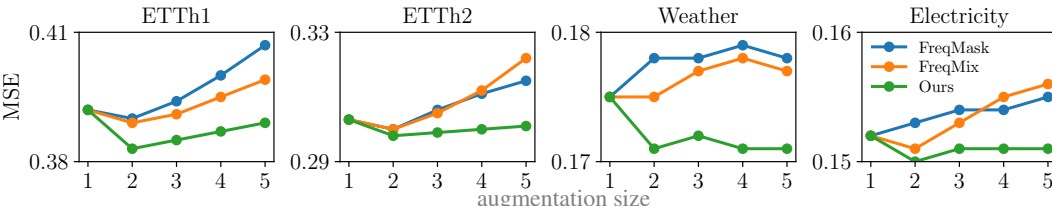

Figure 7: MSE with different augmentation sizes using iTransformer Liu et al. (2024). An augmentation size of two, which was used in previous experiments, achieves the best results in most cases. Our method is more resistant to larger augmentation sizes, indicating the improved augmented-original gap.

As shown in Fig. 7, the performance of FreqMix and FreqMask declines significantly after an augmentation size of two. This is due to the original-augmented gap caused by excessive perturbations. Our method is less sensitive to augmentation size and even performs better with larger augmentation sizes on the Weather dataset. The results in Fig. 7 reveal that our augmented samples are more consistent with the original data, demonstrating less original-augmented gaps.

## 5 CONCLUSION

We proposed the dominant shuffle, a simple yet highly effective data augmentation technique for time series prediction. Our method mitigates the domain gap between augmented and original data by limiting the perturbation to dominant frequencies, and uses shuffles to avoid external noises. Although our method is straightforward and effective, it is primarily based on heuristics and lacks a deep theoretical foundation. Instead of relying on theoretical justifications, we performed extensive experiments across a diverse range of datasets, baseline models, and augmentation methods to validate its consistent improvements under various configurations. Investigating the theoretical justifications and principles of the proposed method presents a promising avenue for future research that could enhance our understanding of its mechanisms.

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

## A    ADDITIONAL EXPERIMENTAL DETAILS

### A.1    DATASETS

We evaluate the performance of different models and different augmentations for long-term forecasting on 8 well-established datasets, including Weather, Traffic, Electricity, Exchange Rate Wu et al. (2021), and ETT datasets (ETTh1, ETTh2, ETTm1, ETTm2) Zhou et al. (2021). Furthermore, we adopt PEMS Chen et al. (2001) datasets for short-term forecasting. We detail the descriptions of the dataset in Tab. 6.

| Dataset | Variates | Prediction length ($T$) | Total Length (Train:Validation:Test) | Frequency | Information |
|---------|----------|-------------------------|--------------------------------------|-----------|-------------|
| ETTh1,ETTh2 | 7 | {96,192,336,720} | (8545, 2,881, 2,881) | Hourly | Temperature |
| ETTm1,ETTm2 | 7 | {96, 192, 336, 720} | (34465, 11521, 11521) | 15min | Temperature |
| Exchange | 8 | {96, 192, 336, 720} | (5120, 665, 1422) | Daily | Economy |
| Weather | 21 | {96,192,336,720} | (36792, 5271, 10540) | 10min | Weather |
| ECL | 321 | {96,192, 336, 720} | (18317, 2633, 5261) | Hourly | Electricity |
| Traffic | 862 | {96, 192, 336, 720} | (12185, 1757, 3509) | Hourly | Transportation |
| PEMS03 | 358 | {12, 24, 36, 48} | (15617, 5135, 5135) | 5min | Traffic network |
| PEMS04 | 307 | {12, 24, 36, 48} | (10172, 3375, 3375) | 5min | Traffic network |
| PEMS07 | 883 | {12, 24, 36, 48} | (16911, 5622, 5622) | 5min | Traffic network |
| PEMS08 | 170 | {12, 24, 36, 48} | (10690, 3548, 3548) | 5min | Traffic network |

Table 6:  Statistics of the eight datasets used in our experiments.

### A.2    IMPLEMENTATION DETAILS

#### A.2.1    REIMPLEMENTATION OTHER METHODS

For ASD, MSB, and upsample, we reproduce them based on the descriptions in their original paper Bandara et al. (2021); Forestier et al. (2017); Semenoglou et al. (2023). For STAug Zhang et al. (2023) and MixMask Chen et al. (2023a), we use their official code. For Robusttad Gao et al. (2020), we reproduce it by adding Gaussian noise to the frequency components of a time series. For FreqAdd Zhang et al. (2022b), we perturb a single low-frequency component by setting its magnitude to half of the maximum magnitude. For FreqPool Chen et al. (2023b), we apply it by maximum pooling of the entire spectrum with size=4. For a fair comparison, all frequency-domain methods target both the data-label pair.

#### A.2.2    DIFFERENT PERTURBATIONS

In our ablation study, we define minor frequencies as other components except for the frequency components with the top 10 magnitudes. In Tab. 4, Mask on the full spectrum is similar to FrAug Chen et al. (2023a). Mask on dominant frequencies means mask within frequency components with the top 10 magnitudes, Mask on minor frequencies is the opposite. In Tab. 5, Noise means adding Gaussian noise to the selected frequency components. For Random, we first get the maximum and minimum magnitude of the selected frequency components and then randomly assigned magnitude within the max-min range.

## B    COMPLEMENTARY RESULTS

### B.1    LONG-TERM PREDICTION ON EIGHT DATASETS

 Tab. 7 to 9 show the full results of the long-term prediction on eight datasets. Our method improves the performance of iTransformer by 13% in Electricity when the predicted length is 720, and it improves the performance of Autoformer by 28% in ETTm1 when the predicted length is 720. Our method also improves the performance of MICN by 18% in ETTh2 when the predicted length is 192 and the performance of SCINet by 21% in Electricity when the predicted length is 720. Similarly, our method improves the performance of Lightts by 29% in ETTh2 when the predicted length is 720

and the performance of TiDE by 24% in Electricity when the predicted length is 192. It is worth noting that the strong baseline MixMask falls short in Exchange rate, whose main goal is to predict trends. But our method improves the performance of Autoformer by 34% in Exchange rate when the predicted length is 720, and it improves the performance of Lightts by 37% in Exchange rate when the predicted length is 96. These results demonstrate the effectiveness of our method for long-term prediction, as it consistently improves the performance of SOTA methods in different datasets.

| Method | ETTh1 96 | 192 | 336 | 720 | ETTh2 96 | 192 | 336 | 720 | ETTm1 96 | 192 | 336 | 720 | ETTm2 96 | 192 | 336 | 720 |
|---|---|---|---|---|---|---|---|---|---|---|---|---|---|---|---|---|
| **iTransformer** | | | | | | | | | | | | | | | | |
| Baseline | 0.392 | 0.447 | 0.483 | 0.516 | 0.303 | 0.381 | 0.412 | 0.434 | 0.344 | 0.383 | 0.421 | 0.494 | 0.183 | 0.251 | 0.311 | 0.412 |
| ASD Forestier et al. (2017) | 0.398 | 0.456 | 0.483 | 0.512 | 0.310 | 0.388 | 0.432 | 0.452 | 0.340 | 0.382 | 0.454 | 0.492 | 0.199 | 0.254 | 0.341 | 0.423 |
| MSB Bandara et al. (2021) | 0.387 | 0.460 | 0.494 | 0.531 | 0.309 | 0.382 | 0.447 | 0.433 | 0.339 | 0.386 | 0.467 | 0.510 | 0.187 | 0.267 | 0.332 | 0.452 |
| Upsample Semenoglou et al. (2023) | 0.391 | 0.445 | 0.481 | 0.519 | 0.305 | 0.381 | 0.419 | 0.430 | 0.351 | 0.381 | 0.432 | 0.489 | 0.196 | 0.279 | 0.320 | 0.411 |
| FreqAdd Zhang et al. (2022b) | 0.389 | 0.446 | 0.475 | 0.510 | 0.300 | 0.384 | 0.416 | 0.438 | 0.350 | 0.385 | 0.422 | 0.490 | 0.187 | 0.253 | 0.311 | 0.415 |
| FreqPool Chen et al. (2023b) | 0.433 | 0.456 | 0.497 | 0.532 | 0.313 | 0.392 | 0.415 | 0.450 | 0.347 | 0.392 | 0.430 | 0.499 | 0.187 | 0.256 | 0.324 | 0.449 |
| Robusttad Gao et al. (2020) | 0.390 | 0.445 | 0.497 | 0.510 | 0.312 | 0.388 | 0.412 | 0.439 | 0.353 | 0.382 | 0.421 | 0.498 | 0.189 | 0.255 | 0.309 | 0.428 |
| STAug Zhang et al. (2023) | 0.390 | 0.445 | 0.489 | 0.511 | 0.323 | 0.428 | 0.486 | 0.483 | 0.339 | 0.383 | **0.417** | 0.485 | 0.196 | 0.267 | 0.339 | 0.449 |
| MixMask Chen et al. (2023a) | 0.388 | 0.440 | 0.477 | 0.504 | 0.301 | **0.380** | 0.414 | 0.434 | 0.334 | 0.375 | 0.421 | **0.485** | **0.178** | 0.248 | 0.311 | **0.407** |
| Ours | **0.383** | **0.438** | **0.473** | **0.492** | **0.298** | 0.382 | **0.411** | **0.428** | **0.332** | **0.374** | 0.424 | 0.492 | **0.178** | **0.246** | **0.309** | 0.409 |
| **AutoFormer** | | | | | | | | | | | | | | | | |
| Baseline | 0.429 | 0.440 | 0.495 | 0.498 | 0.381 | 0.443 | 0.471 | 0.475 | 0.467 | 0.610 | 0.529 | 0.773 | 0.233 | 0.278 | 0.383 | 0.488 |
| ASD | 0.450 | 0.485 | 0.523 | 0.556 | 0.370 | 0.465 | 0.476 | 0.503 | 0.480 | 0.620 | 0.502 | 0.633 | 0.231 | 0.282 | 0.379 | 0.499 |
| MSB | 0.462 | 0.517 | 0.612 | 0.579 | 0.434 | 0.523 | 0.556 | 0.462 | 0.499 | 0.645 | 0.553 | 0.721 | 0.232 | 0.285 | 0.389 | 0.487 |
| Upsample | 0.416 | 0.523 | 0.480 | **0.482** | 0.353 | 0.460 | 0.455 | 0.509 | 0.498 | 0.630 | 0.512 | 0.667 | 0.234 | 0.291 | 0.382 | 0.521 |
| FreqAdd | 0.460 | 0.487 | 0.497 | 0.525 | 0.367 | 0.439 | 0.480 | 0.504 | 0.419 | 0.554 | 0.546 | 0.569 | 0.223 | 0.268 | 0.330 | 0.458 |
| FreqPool | 0.446 | 0.457 | 0.523 | 0.512 | 0.392 | 0.442 | 0.470 | 0.493 | 0.479 | 0.623 | 0.510 | 0.754 | 0.250 | 0.291 | 0.394 | 0.482 |
| Robusttad | 0.437 | 0.452 | 0.492 | 0.477 | 0.367 | 0.497 | 0.502 | 0.527 | 0.432 | 0.510 | 0.553 | 0.623 | 0.235 | 0.291 | 0.375 | 0.478 |
| STAug | 0.429 | 0.478 | 0.505 | 0.506 | 0.354 | 0.443 | 0.496 | 0.495 | 0.415 | 0.581 | 0.588 | 0.693 | 0.224 | 0.291 | 0.338 | 0.431 |
| MixMask | 0.420 | 0.445 | 0.467 | 0.474 | 0.358 | 0.421 | 0.410 | 0.467 | 0.415 | 0.510 | 0.491 | 0.588 | 0.211 | 0.267 | 0.340 | 0.451 |
| Ours | **0.409** | **0.436** | **0.458** | 0.486 | **0.335** | **0.419** | **0.453** | **0.452** | **0.392** | **0.506** | **0.491** | **0.559** | **0.210** | **0.266** | **0.329** | **0.429** |
| **MICN** | | | | | | | | | | | | | | | | |
| Baseline | 0.384 | 0.425 | 0.464 | 0.574 | 0.358 | 0.518 | 0.566 | 0.827 | 0.313 | 0.360 | 0.389 | 0.461 | 0.200 | 0.282 | 0.375 | 0.606 |
| ASD | 0.380 | 0.430 | 0.472 | 0.523 | 0.377 | 0.539 | 0.620 | 0.843 | 0.315 | 0.362 | 0.399 | 0.457 | 0.189 | 0.331 | 0.399 | 0.617 |
| MSB | 0.423 | 0.423 | 0.501 | 0.559 | 0.402 | 0.623 | 0.790 | 1.126 | 0.330 | 0.358 | 0.402 | 0.459 | 0.192 | 0.279 | 0.376 | 0.651 |
| Upsample | 0.396 | 0.435 | 0.463 | 0.550 | 0.366 | 0.500 | 0.831 | 0.752 | 0.339 | 0.377 | 0.402 | 0.475 | 0.203 | 0.291 | 0.372 | 0.595 |
| FreqAdd | 0.390 | 0.430 | 0.477 | 0.643 | 0.370 | 0.521 | 0.626 | 0.975 | 0.316 | 0.360 | 0.407 | 0.478 | 0.176 | 0.273 | 0.378 | 0.614 |
| FreqPool | 0.399 | 0.465 | 0.473 | 0.572 | 0.365 | 0.553 | 0.550 | 0.812 | 0.336 | 0.372 | 0.397 | 0.466 | 0.212 | 0.287 | 0.390 | 0.623 |
| Robusttad | 0.392 | 0.436 | 0.491 | 0.556 | 0.339 | 0.529 | 0.553 | 0.998 | 0.339 | 0.359 | 0.396 | 0.472 | 0.200 | 0.296 | 0.356 | 0.617 |
| STAug | 0.374 | 0.429 | 0.489 | 0.608 | 0.413 | 0.760 | 1.330 | 2.608 | 0.313 | 0.360 | 0.418 | 0.483 | 0.180 | 0.264 | 0.323 | 0.670 |
| MixMask | 0.378 | 0.423 | 0.461 | 0.521 | 0.339 | 0.488 | 0.544 | 0.735 | **0.301** | **0.352** | 0.401 | **0.454** | 0.183 | 0.278 | 0.356 | 0.528 |
| Ours | **0.373** | **0.421** | **0.452** | **0.510** | **0.310** | **0.427** | **0.507** | **0.731** | 0.314 | 0.360 | **0.387** | 0.470 | **0.174** | **0.263** | **0.346** | **0.502** |
| **SCINet** | | | | | | | | | | | | | | | | |
| Baseline | 0.485 | 0.506 | 0.519 | 0.552 | 0.372 | 0.416 | 0.429 | 0.470 | 0.316 | 0.353 | 0.387 | 0.431 | 0.184 | 0.240 | 0.295 | 0.385 |
| ASD | 0.494 | 0.480 | 0.491 | 0.559 | 0.362 | 0.402 | 0.432 | 0.499 | 0.331 | 0.367 | 0.389 | 0.453 | 0.197 | 0.238 | 0.296 | 0.432 |
| MSB | 0.489 | 0.466 | 0.502 | 0.547 | 0.359 | 0.396 | 0.458 | 0.476 | 0.320 | 0.351 | 0.396 | 0.478 | 0.182 | 0.237 | 0.289 | 0.449 |
| Upsample | 0.471 | 0.457 | 0.479 | 0.541 | 0.379 | 0.407 | 0.403 | 0.482 | 0.342 | 0.386 | 0.399 | 0.442 | 0.179 | 0.254 | 0.292 | 0.401 |
| FreqAdd | 0.428 | 0.452 | 0.469 | 0.532 | 0.335 | 0.385 | 0.403 | 0.447 | 0.304 | 0.338 | 0.373 | 0.421 | 0.174 | 0.228 | 0.286 | 0.380 |
| FreqPool | 0.499 | 0.510 | 0.557 | 0.549 | 0.410 | 0.453 | 0.432 | 0.475 | 0.331 | 0.362 | 0.379 | 0.432 | 0.185 | 0.239 | 0.302 | 0.399 |
| Robusttad | 0.462 | 0.501 | 0.498 | 0.559 | 0.362 | 0.431 | 0.419 | 0.496 | 0.331 | 0.351 | 0.394 | 0.438 | 0.182 | 0.247 | 0.299 | 0.402 |
| STAug | 0.457 | 0.500 | 0.524 | 0.534 | 0.538 | 0.636 | 0.681 | 0.648 | 0.319 | 0.357 | 0.389 | 0.445 | 0.323 | 0.407 | 0.514 | 0.668 |
| MixMask | 0.427 | 0.452 | 0.465 | 0.548 | **0.335** | 0.377 | 0.400 | 0.438 | **0.302** | 0.341 | 0.376 | 0.423 | **0.174** | 0.230 | 0.289 | **0.368** |
| Ours | **0.417** | **0.443** | **0.461** | **0.527** | **0.335** | **0.375** | **0.392** | **0.421** | **0.302** | **0.338** | **0.372** | **0.420** | **0.174** | **0.228** | **0.283** | 0.372 |
| **TiDE** | | | | | | | | | | | | | | | | |
| Baseline | 0.401 | 0.434 | 0.521 | 0.558 | 0.304 | 0.350 | 0.331 | 0.399 | 0.311 | 0.340 | 0.366 | 0.420 | 0.166 | 0.220 | 0.273 | 0.356 |
| ASD | 0.417 | 0.441 | 0.513 | 0.556 | 0.320 | 0.351 | 0.360 | 0.422 | 0.319 | 0.341 | 0.399 | 0.432 | 0.177 | 0.241 | 0.291 | 0.371 |
| MSB | 0.422 | 0.476 | 0.529 | 0.579 | 0.331 | 0.379 | 0.334 | 0.401 | 0.302 | 0.356 | 0.382 | 0.451 | 0.182 | 0.232 | 0.287 | 0.359 |
| Upsample | 0.431 | 0.452 | 0.533 | 0.604 | 0.346 | 0.372 | 0.350 | 0.456 | 0.324 | 0.339 | 0.378 | 0.463 | 0.203 | 0.246 | 0.306 | 0.366 |
| FreqAdd | 0.385 | 0.420 | 0.477 | 0.505 | 0.289 | 0.336 | 0.330 | 0.390 | 0.309 | 0.339 | 0.365 | 0.417 | 0.164 | 0.219 | 0.273 | 0.355 |
| FreqPool | 0.423 | 0.455 | 0.510 | 0.592 | 0.312 | .376 | 0.339 | 0.397 | 0.319 | 0.352 | 0.397 | 0.453 | 0.179 | 0.231 | 0.299 | 0.371 |
| Robusttad | 0.396 | 0.432 | 0.521 | 0.537 | 0.331 | 0.352 | 0.337 | 0.398 | 0.321 | 0.346 | 0.382 | 0.437 | 0.180 | 0.225 | 0.282 | 0.371 |
| STAug | 0.515 | 0.535 | 0.521 | 0.558 | 0.390 | 0.437 | 0.403 | 0.508 | 0.310 | 0.337 | **0.364** | 0.417 | 0.222 | 0.343 | 0.515 | 0.847 |
| MixMask | **0.385** | 0.420 | 0.478 | 0.507 | 0.289 | 0.339 | 0.336 | 0.391 | 0.299 | 0.332 | 0.367 | 0.416 | **0.165** | 0.219 | **0.271** | 0.347 |
| Ours | **0.385** | **0.414** | **0.467** | **0.498** | **0.283** | **0.332** | **0.324** | **0.388** | **0.297** | **0.328** | 0.365 | **0.412** | **0.165** | **0.218** | **0.271** | 0.350 |
| **LightTS** | | | | | | | | | | | | | | | | |
| Baseline | 0.448 | 0.444 | 0.663 | 0.706 | 0.369 | 0.476 | 0.738 | 1.165 | 0.323 | 0.347 | 0.428 | 0.476 | 0.212 | 0.237 | 0.350 | 0.473 |
| ASD | 0.451 | 0.476 | 0.633 | 0.681 | 0.392 | 0.469 | 0.701 | 0.998 | 0.356 | 0.352 | 0.441 | 0.478 | 0.258 | 0.251 | 0.351 | 0.483 |
| MSB | 0.467 | 0.463 | 0.627 | 0.652 | 0.378 | 0.472 | 0.652 | 1.123 | 0.371 | 0.349 | 0.430 | 0.479 | 0.236 | 0.242 | 0.359 | 0.471 |
| Upsample | 0.449 | 0.472 | 0.610 | 0.637 | 0.401 | 0.487 | 0.714 | 1.245 | 0.329 | 0.366 | 0.453 | 0.492 | 0.241 | 0.255 | 0.366 | 0.492 |
| FreqAdd | 0.417 | 0.430 | 0.578 | 0.622 | 0.351 | 0.453 | 0.689 | 1.125 | 0.322 | 0.352 | 0.400 | 0.450 | 0.206 | 0.237 | 0.327 | 0.455 |
| FreqPool | 0.463 | 0.471 | 0.652 | 0.690 | 0.369 | 0.512 | 0.723 | 1.264 | 0.336 | 0.351 | 0.442 | 0.497 | 0.233 | 0.259 | 0.372 | 0.453 |
| Robusttad | 0.445 | 0.442 | 0.590 | 0.654 | 0.372 | 0.468 | 0.699 | 0.982 | 0.331 | 0.352 | 0.441 | 0.462 | 0.232 | 0.227 | 0.342 | 0.446 |
| STAug | 0.445 | 0.441 | 0.669 | 0.714 | 0.520 | 0.807 | 2.101 | 2.467 | 0.320 | 0.343 | 0.427 | 0.476 | 0.230 | 0.266 | 0.372 | 0.475 |
| MixMask | 0.417 | 0.429 | 0.575 | 0.620 | 0.337 | 0.426 | 0.643 | 0.993 | **0.316** | **0.340** | 0.398 | 0.447 | 0.199 | **0.233** | 0.322 | 0.440 |
| Ours | **0.405** | **0.423** | **0.565** | **0.603** | **0.335** | **0.395** | **0.575** | **0.827** | 0.322 | **0.340** | **0.391** | **0.440** | **0.195** | 0.245 | **0.312** | **0.422** |

Table 7: MSE of the long-term prediction on the ETT Zhou et al. (2021) datasets.

## B.2 Long-term Prediction with PatchTST, Pathformer, and PDF

Tab. 10 and 11 demonstrate the experiments results on three models: PatchTST Nie et al. (2023b), Pathformer Chen et al. (2024), and PDF Dai et al. (2024). From the experimental results, it can be seen that our method still achieved the best results compared to other DA methods.

## B.3 Example predictions

We provided example prediction results on different datasets in Fig. 8

| Model | Method | Eletricity 96 | 192 | 336 | 720 | Weather 96 | 192 | 336 | 720 | Exchange Rate 96 | 192 | 336 | 720 | Traffic 96 | 192 | 336 | 720 |
|---|---|---|---|---|---|---|---|---|---|---|---|---|---|---|---|---|---|
| iTransformer | Baseline | 0.152 | 0.159 | 0.179 | 0.230 | 0.175 | 0.224 | 0.228 | 0.281 | 0.086 | 0.180 | 0.335 | 0.856 | 0.399 | 0.418 | 0.428 | 0.463 |
| | ASD Forestier et al. (2017) | 0.173 | 0.179 | 0.201 | 0.234 | 0.191 | 0.223 | 0.280 | 0.364 | 0.088 | 0.183 | 0.343 | 0.872 | 0.431 | 0.428 | 0.430 | 0.478 |
| | MSB Bandara et al. (2021) | 0.182 | 0.182 | 0.194 | 0.267 | 0.185 | 0.235 | 0.284 | 0.359 | 0.089 | 0.189 | 0.359 | 0.907 | 0.417 | 0.416 | 0.422 | 0.471 |
| | Upsample Semenoglou et al. (2023) | 0.166 | 0.188 | 0.216 | 0.221 | 0.204 | 0.257 | 0.291 | 0.373 | 0.086 | 0.180 | 0.338 | 0.834 | 0.433 | 0.419 | 0.433 | 0.476 |
| | FreqAdd Zhang et al. (2022b) | 0.150 | 0.157 | 0.172 | 0.204 | 0.181 | 0.230 | 0.285 | 0.362 | 0.087 | 0.181 | 0.333 | 0.837 | 0.480 | 0.441 | 0.450 | 0.501 |
| | FreqPool Chen et al. (2023b) | 0.169 | 0.170 | 0.194 | 0.237 | 0.184 | 0.223 | 0.279 | 0.378 | 0.088 | 0.183 | 0.330 | 0.832 | 0.410 | 0.429 | 0.433 | 0.476 |
| | Robusttad Gao et al. (2020) | 0.150 | 0.157 | 0.176 | 0.210 | 0.172 | 0.225 | 0.281 | 0.357 | 0.087 | 0.179 | 0.329 | 0.833 | 0.406 | 0.417 | 0.429 | 0.458 |
| | STAug Zhang et al. (2023) | 0.160 | 0.173 | 0.218 | 0.372 | 0.206 | 0.264 | 0.319 | 0.385 | 0.086 | 0.178 | 0.335 | 0.866 | 0.413 | 0.432 | 0.449 | 0.481 |
| | MixMask Chen et al. (2023a) | 0.151 | 0.158 | 0.173 | 0.205 | 0.175 | 0.224 | 0.279 | 0.354 | 0.089 | 0.178 | 0.328 | 0.845 | 0.395 | **0.401** | **0.418** | 0.450 |
| | Ours | **0.150** | **0.156** | **0.171** | **0.199** | **0.171** | **0.221** | **0.276** | **0.351** | **0.086** | **0.176** | **0.313** | **0.821** | **0.394** | 0.412 | 0.423 | **0.448** |
| AutoFormer | Baseline | 0.203 | 0.208 | 0.231 | 0.239 | 0.241 | 0.314 | 0.341 | 0.425 | 0.143 | 0.305 | 0.470 | 1.056 | 0.640 | 0.645 | 0.611 | 0.658 |
| | ASD | 0.247 | 0.216 | 0.221 | 0.235 | 0.652 | 0.392 | 0.416 | 0.513 | 0.141 | 0.280 | 0.579 | 1.240 | 0.631 | 0.602 | 0.607 | 0.643 |
| | MSB | 0.237 | 0.256 | 0.295 | 0.236 | 0.256 | 0.379 | 0.402 | 0.468 | 0.156 | 0.254 | 0.513 | 1.339 | 0.652 | 0.665 | 0.643 | 0.65 |
| | Upsample | 0.201 | 0.209 | 0.232 | 0.268 | 0.281 | 0.294 | 0.329 | 0.385 | 0.141 | 0.292 | 0.553 | 1.295 | 0.653 | 0.676 | 0.702 | 0.694 |
| | FreqAdd | 0.193 | 0.197 | 0.212 | 0.225 | 0.255 | 0.323 | 0.370 | 0.419 | 0.143 | 0.369 | 0.716 | 1.173 | 0.613 | 0.598 | 0.617 | 0.639 |
| | FreqPool | 0.213 | 0.224 | 0.234 | 0.257 | 0.237 | 0.339 | 0.372 | 0.446 | 0.142 | 0.336 | 0.532 | 1.014 | 0.63 | 0.598 | 0.603 | 0.639 |
| | Robusttad | 0.230 | 0.242 | 0.261 | 0.231 | 0.27 | 0.334 | 0.351 | 0.429 | 0.142 | 0.309 | 0.462 | 1.123 | 0.621 | 0.614 | 0.612 | 0.646 |
| | STAug | 0.191 | 0.206 | 0.217 | 0.234 | 0.250 | 0.300 | 0.347 | 0.418 | 0.140 | 0.326 | 0.594 | 1.176 | 0.632 | 0.619 | 0.632 | 0.640 |
| | MixMask | 0.177 | 0.194 | 0.206 | 0.224 | 0.240 | 0.302 | 0.330 | 0.422 | 0.141 | 0.284 | 0.453 | 0.778 | **0.560** | 0.584 | 0.594 | **0.635** |
| | Ours | **0.171** | **0.191** | **0.203** | **0.219** | **0.214** | **0.273** | **0.327** | **0.383** | **0.136** | **0.243** | **0.418** | **0.695** | 0.577 | **0.581** | **0.592** | 0.638 |
| MICN | Baseline | 0.171 | 0.183 | 0.198 | 0.224 | 0.188 | 0.241 | 0.278 | 0.350 | 0.091 | 0.185 | 0.355 | 0.941 | 0.522 | 0.540 | 0.553 | 0.573 |
| | ASD | 0.165 | 0.174 | 0.190 | 0.237 | 0.189 | 0.242 | 0.276 | 0.354 | 0.087 | 0.175 | 0.337 | 1.203 | 0.505 | 0.534 | 0.541 | 0.539 |
| | MSB | 0.179 | 0.182 | 0.201 | 0.225 | 0.201 | 0.250 | 0.291 | 0.365 | 0.088 | 0.176 | 0.360 | 0.995 | 0.513 | 0.532 | 0.528 | 0.556 |
| | Upsample | 0.182 | 0.180 | 0.203 | 0.220 | 0.193 | 0.249 | 0.279 | 0.372 | 0.084 | 0.171 | 0.313 | **0.702** | 0.533 | 0.559 | 0.556 | 0.590 |
| | FreqAdd | 0.160 | 0.169 | 0.182 | 0.199 | 0.180 | 0.234 | 0.282 | 0.350 | 0.087 | 0.174 | 0.349 | 0.923 | 0.503 | 0.527 | 0.520 | 0.571 |
| | FreqPool | 0.182 | 0.203 | 0.241 | 0.256 | 0.192 | 0.257 | 0.278 | 0.351 | 0.089 | 0.179 | 0.394 | 0.923 | 0.531 | 0.539 | 0.556 | 0.592 |
| | Robusttad | 0.179 | 0.220 | 0.234 | 0.227 | 0.192 | 0.239 | 0.292 | 0.343 | 0.085 | 0.179 | 0.336 | 0.932 | 0.510 | 0.532 | 0.547 | 0.597 |
| | STAug | 0.180 | 0.195 | 0.210 | 0.224 | 0.272 | 0.356 | 0.433 | 0.559 | 0.092 | 0.183 | 0.313 | 0.790 | 0.512 | 0.533 | 0.529 | 0.585 |
| | MixMask | 0.159 | **0.165** | **0.178** | **0.195** | 0.185 | 0.239 | 0.281 | 0.344 | 0.086 | 0.174 | 0.337 | 0.796 | 0.490 | 0.512 | 0.519 | **0.538** |
| | Ours | **0.157** | 0.168 | **0.178** | 0.211 | **0.179** | **0.232** | **0.275** | **0.342** | **0.084** | **0.169** | **0.303** | 0.750 | 0.501 | **0.507** | **0.518** | 0.556 |
| SCINet | Baseline | 0.212 | 0.237 | 0.255 | 0.286 | 0.229 | 0.282 | 0.334 | 0.402 | 0.099 | 0.191 | 0.356 | 0.916 | 0.550 | 0.526 | 0.545 | 0.596 |
| | ASD | 0.229 | 0.241 | 0.239 | 0.282 | 0.254 | 0.276 | 0.356 | 0.462 | 0.095 | 0.204 | 0.379 | 1.230 | 0.537 | 0.521 | 0.541 | 0.570 |
| | MSB | 0.232 | 0.237 | 0.228 | 0.274 | 0.279 | 0.265 | 0.374 | 0.454 | 0.093 | 0.267 | 0.402 | 0.965 | 0.520 | 0.510 | 0.537 | 0.565 |
| | Upsample | 0.250 | 0.232 | 0.271 | 0.309 | 0.243 | 0.299 | 0.361 | 0.431 | 0.092 | 0.196 | **0.311** | 0.932 | 0.519 | 0.536 | 0.528 | 0.576 |
| | FreqAdd | 0.176 | 0.195 | 0.212 | 0.237 | 0.208 | 0.258 | 0.309 | 0.385 | 0.092 | 0.186 | 0.343 | 0.920 | **0.492** | 0.497 | 0.512 | 0.550 |
| | FreqPool | 0.230 | 0.221 | 0.242 | 0.339 | 0.261 | 0.290 | 0.337 | 0.456 | 0.096 | 0.183 | 0.551 | 0.938 | 0.557 | 0.519 | 0.533 | 0.562 |
| | Robusttad | 0.189 | 0.202 | 0.210 | 0.243 | 0.229 | 0.281 | 0.331 | 0.410 | 0.093 | 0.186 | 0.334 | 0.957 | 0.523 | 0.519 | 0.522 | 0.569 |
| | STAug | 0.210 | 0.239 | 0.282 | 0.411 | 0.277 | 0.329 | 0.372 | 0.435 | 0.098 | 0.191 | 0.342 | 0.931 | 0.560 | 0.517 | 0.521 | 0.566 |
| | MixMask | **0.171** | **0.188** | 0.204 | 0.230 | 0.205 | 0.250 | 0.310 | **0.374** | 0.093 | 0.179 | 0.336 | 0.928 | 0.495 | **0.492** | 0.511 | 0.551 |
| | Ours | 0.172 | **0.188** | **0.200** | **0.225** | **0.197** | **0.246** | **0.299** | 0.379 | **0.091** | **0.175** | 0.342 | **0.890** | 0.500 | 0.495 | **0.509** | **0.544** |
| TiDE | Baseline | 0.207 | 0.197 | 0.211 | 0.238 | 0.177 | 0.220 | 0.265 | 0.323 | 0.093 | 0.184 | 0.330 | 0.860 | 0.452 | 0.450 | 0.451 | 0.479 |
| | ASD | 0.232 | 0.220 | 0.231 | 0.265 | 0.189 | 0.221 | 0.297 | 0.332 | 0.095 | 0.206 | 0.351 | 0.962 | 0.477 | 0.462 | 0.450 | 0.506 |
| | MSB | 0.210 | 0.219 | 0.253 | 0.261 | 0.199 | 0.254 | 0.273 | 0.339 | 0.092 | 0.179 | 0.358 | 0.941 | 0.461 | 0.451 | 0.455 | 0.510 |
| | Upsample | 0.206 | 0.199 | 0.223 | 0.274 | 0.203 | 0.267 | 0.331 | 0.355 | 0.091 | 0.182 | 0.331 | 0.852 | 0.490 | 0.466 | 0.472 | 0.493 |
| | FreqAdd | 0.150 | 0.163 | 0.177 | 0.209 | **0.173** | **0.216** | 0.263 | **0.322** | **0.088** | 0.180 | 0.330 | 0.848 | 0.429 | 0.441 | 0.440 | 0.471 |
| | FreqPool | 0.224 | 0.238 | 0.233 | 0.270 | 0.189 | 0.224 | 0.292 | 0.334 | 0.092 | 0.334 | 0.521 | 1.124 | 0.453 | 0.466 | 0.479 | 0.503 |
| | Robusttad | 0.176 | 0.166 | 0.182 | 0.229 | 0.182 | 0.231 | 0.279 | 0.330 | 0.099 | 0.232 | 0.331 | 0.924 | 0.449 | 0.430 | 0.438 | 0.482 |
| | STAug | 0.230 | 0.210 | 0.192 | 0.225 | 0.205 | 0.247 | 0.292 | 0.364 | 0.092 | 0.184 | 0.330 | 0.859 | 0.466 | 0.455 | 0.471 | 0.480 |
| | MixMask | **0.143** | 0.155 | **0.164** | 0.210 | **0.173** | **0.216** | 0.263 | 0.323 | 0.089 | 0.180 | 0.329 | 0.861 | **0.421** | 0.427 | 0.434 | **0.466** |
| | Ours | **0.143** | **0.150** | 0.165 | **0.202** | 0.177 | 0.219 | **0.261** | **0.322** | **0.088** | **0.179** | **0.324** | **0.847** | 0.423 | **0.426** | **0.433** | **0.466** |
| LightTS | Baseline | 0.210 | 0.169 | 0.182 | 0.212 | 0.168 | 0.210 | 0.260 | 0.320 | 0.139 | 0.252 | 0.412 | 0.840 | 0.505 | 0.515 | 0.539 | 0.587 |
| | ASD | 0.225 | 0.179 | 0.198 | 0.232 | 0.179 | 0.21 | 0.271 | 0.321 | 0.132 | 0.320 | 0.436 | 1.036 | 0.510 | 0.514 | 0.534 | 0.579 |
| | MSB | 0.233 | 0.182 | 0.204 | 0.228 | 0.170 | 0.214 | 0.259 | 0.332 | 0.117 | 0.294 | 0.502 | 0.964 | 0.532 | 0.510 | 0.539 | 0.584 |
| | Upsample | 0.246 | 0.179 | 0.211 | 0.254 | 0.182 | 0.223 | 0.257 | 0.336 | 0.099 | 0.251 | 0.369 | 0.702 | 0.522 | 0.547 | 0.532 | 0.597 |
| | FreqAdd | 0.213 | 0.159 | 0.177 | 0.210 | 0.164 | 0.207 | 0.258 | 0.317 | 0.098 | 0.522 | 0.565 | 1.583 | 0.492 | 0.500 | 0.530 | 0.572 |
| | FreqPool | 0.219 | 0.174 | 0.197 | 0.236 | 0.193 | 0.254 | 0.267 | 0.339 | 0.099 | 0.275 | 0.394 | 0.793 | 0.501 | 0.519 | 0.533 | 0.592 |
| | Robusttad | 0.212 | 0.169 | 0.181 | 0.223 | 0.172 | 0.223 | 0.259 | 0.324 | 0.092 | 0.279 | 0.451 | 0.796 | 0.499 | 0.502 | 0.521 | 0.572 |
| | STAug | 0.224 | 0.267 | 0.294 | 0.351 | 0.214 | 0.263 | 0.382 | 0.371 | 0.096 | **0.212** | 0.380 | 0.690 | 0.520 | 0.534 | 0.520 | 0.596 |
| | MixMask | **0.192** | 0.158 | 0.175 | 0.211 | **0.163** | 0.206 | 0.257 | 0.318 | 0.099 | 0.384 | 0.518 | 0.774 | 0.486 | 0.499 | 0.517 | **0.555** |
| | Ours | 0.210 | **0.156** | **0.173** | **0.206** | 0.165 | **0.205** | **0.249** | **0.312** | **0.088** | 0.243 | **0.361** | **0.676** | **0.483** | **0.497** | **0.515** | 0.567 |

Table 8: MSE of the long-term prediction on the Electricity, traffic, Weather, and Exchange Rate Wu et al. (2021) datasets.

| Methods | PEMS03 12 | 24 | 36 | 48 | PEMS04 12 | 24 | 36 | 48 | PEMS07 12 | 24 | 36 | 48 | PEMS08 12 | 24 | 36 | 48 |
|---|---|---|---|---|---|---|---|---|---|---|---|---|---|---|---|---|
| Baseline | 0.070 | 0.097 | 0.134 | 0.164 | 0.088 | 0.124 | 0.160 | 0.196 | 0.067 | 0.097 | 0.128 | 0.156 | 0.088 | 0.136 | 0.191 | 0.248 |
| ASD Forestier et al. (2017) | 0.072 | 0.096 | 0.152 | 0.239 | 0.098 | 0.132 | 0.156 | 0.190 | 0.069 | 0.099 | 0.154 | 0.181 | 0.089 | 0.138 | 0.196 | 0.247 |
| MSB Bandara et al. (2021) | 0.096 | 0.131 | 0.129 | 0.214 | 0.087 | 0.134 | 0.167 | 0.219 | 0.098 | 0.096 | 0.137 | 0.165 | 0.096 | 0.137 | 0.210 | 0.256 |
| Upsample Semenoglou et al. (2023) | 0.069 | 0.096 | 0.128 | 0.179 | 0.087 | 0.124 | 0.158 | 0.199 | 0.072 | 0.099 | 0.127 | 0.155 | 0.088 | 0.140 | 0.192 | 0.245 |
| FreqAdd Zhang et al. (2022b) | 1.036 | 0.104 | 0.251 | 0.362 | 0.088 | 0.125 | 0.159 | 0.201 | 0.067 | 0.097 | 0.127 | 0.155 | 0.089 | 0.135 | 0.192 | 0.253 |
| FreqPool Chen et al. (2023b) | 1.234 | 0.178 | 0.296 | 0.451 | 0.099 | 0.145 | 0.178 | 0.226 | 0.079 | 0.104 | 0.152 | 0.172 | 0.099 | 0.155 | 0.203 | 0.264 |
| Robusttad Gao et al. (2020) | 0.082 | 0.098 | 0.132 | 1.520 | 0.089 | 0.123 | 0.161 | 0.195 | 0.067 | 0.097 | 0.129 | 0.157 | 0.092 | 0.135 | 0.189 | 0.26 |
| STAug Zhang et al. (2023) | 0.079 | 0.112 | 0.195 | 0.456 | 0.087 | 0.120 | 0.162 | 0.304 | 0.066 | 0.096 | 0.132 | 0.165 | 0.092 | 0.147 | 0.192 | 0.276 |
| Mask Chen et al. (2023a) | 0.443 | 1.205 | 0.233 | 1.510 | 0.086 | 0.119 | 0.158 | 0.346 | 0.065 | 0.095 | 0.125 | 0.156 | 0.089 | 0.131 | 0.186 | 0.239 |
| Mix Chen et al. (2023a) | 1.018 | 0.097 | 0.877 | 1.501 | **0.085** | 0.119 | 0.154 | 0.205 | **0.065** | **0.094** | 0.134 | 0.152 | 0.089 | **0.131** | 0.184 | **0.234** |
| Ours | **0.067** | **0.095** | **0.126** | 0.235 | **0.085** | **0.118** | **0.149** | **0.182** | **0.065** | **0.094** | **0.123** | **0.148** | **0.087** | 0.134 | **0.184** | 0.240 |

Table 9: MSE of the Short-term prediction using the iTransformer Liu et al. (2024) on the PEMS datasets Chen et al. (2001).

## B.4 OPTIMAL $k$

We provide the optimal $k$ for all long-term prediction datasets using iTranformer Liu et al. (2024) in Tab. 12 and 13. As can be seen from the table, our method does not need too much effort to find the optimal parameters.

Table 10: Mean Squared Error (MSE) of the long-term prediction on the ETT datasets.

| | Method | ETTh1 | | | | ETTh2 | | | | ETTm1 | | | |
|---|---|---|---|---|---|---|---|---|---|---|---|---|---|
| | | 96 | 192 | 336 | 720 | 96 | 192 | 336 | 720 | 96 | 192 | 336 | 720 |
| PatchTST | original paper | 0.375 | 0.414 | 0.430 | 0.449 | 0.274 | 0.339 | 0.331 | 0.379 | 0.290 | 0.332 | 0.366 | 0.420 |
| | our reproduce | 0.374 | 0.416 | 0.428 | 0.452 | 0.278 | 0.337 | 0.382 | 0.382 | 0.298 | 0.338 | 0.366 | 0.421 |
| | FreqMask | 0.374 | 0.412 | 0.426 | 0.453 | 0.277 | 0.330 | 0.357 | 0.386 | 0.300 | 0.339 | 0.361 | 0.417 |
| | FreqMix | 0.372 | 0.411 | 0.423 | 0.451 | 0.231 | 0.332 | 0.357 | 0.379 | 0.301 | 0.343 | 0.365 | 0.412 |
| | Ours | 0.368 | 0.406 | 0.423 | 0.445 | 0.274 | 0.324 | 0.355 | 0.376 | 0.302 | 0.338 | 0.358 | 0.408 |
| PathFormer | original paper | 0.382 | 0.440 | 0.454 | 0.479 | 0.279 | 0.349 | 0.348 | 0.398 | 0.316 | 0.366 | 0.386 | 0.460 |
| | our reproduce | 0.382 | 0.441 | 0.453 | 0.488 | 0.286 | 0.353 | 0.348 | 0.397 | 0.320 | 0.366 | 0.385 | 0.465 |
| | FreqMask | 0.375 | 0.438 | 0.455 | 0.476 | 0.286 | 0.348 | 0.355 | 0.395 | 0.312 | 0.359 | 0.388 | 0.463 |
| | FreqMix | 0.376 | 0.438 | 0.452 | 0.479 | 0.290 | 0.343 | 0.351 | 0.393 | 0.316 | 0.359 | 0.379 | 0.467 |
| | Ours | 0.370 | 0.432 | 0.449 | 0.470 | 0.286 | 0.341 | 0.355 | 0.389 | 0.312 | 0.357 | 0.377 | 0.459 |
| PDF | original paper | 0.357 | 0.397 | 0.409 | 0.432 | 0.272 | 0.335 | 0.325 | 0.375 | 0.280 | 0.317 | 0.354 | 0.405 |
| | our reproduce | 0.361 | 0.398 | 0.409 | 0.431 | 0.272 | 0.335 | 0.324 | 0.376 | 0.282 | 0.321 | 0.354 | 0.407 |
| | FreqMask | 0.359 | 0.399 | 0.410 | 0.431 | 0.271 | 0.336 | 0.325 | 0.383 | 0.288 | 0.320 | 0.361 | 0.407 |
| | FreqMix | 0.362 | 0.395 | 0.407 | 0.432 | 0.270 | 0.336 | 0.322 | 0.379 | 0.282 | 0.321 | 0.356 | 0.409 |
| | Ours | 0.356 | 0.395 | 0.405 | 0.428 | 0.273 | 0.334 | 0.321 | 0.373 | 0.282 | 0.319 | 0.352 | 0.407 |

Table 11: MSE of the long-term prediction on the ETTh1, ETTh2, and Exchange datasets.

| | Method | Eletricity | | | | Weather | | | | Exchange | | | |
|---|---|---|---|---|---|---|---|---|---|---|---|---|---|
| | | 96 | 192 | 336 | 720 | 96 | 192 | 336 | 720 | 96 | 192 | 336 | 720 |
| PatchTST | original paper | 0.130 | 0.148 | 0.167 | 0.202 | 0.152 | 0.197 | 0.249 | 0.320 | - | - | - | - |
| | our reproduce | 0.138 | 0.148 | 0.163 | 0.197 | 0.152 | 0.196 | 0.249 | 0.320 | 0.087 | 0.188 | 0.311 | 0.798 |
| | FreqMask | 0.138 | 0.146 | 0.163 | 0.189 | 0.155 | 0.200 | 0.248 | 0.319 | 0.088 | 0.189 | 0.335 | 0.815 |
| | FreqMix | 0.141 | 0.149 | 0.162 | 0.192 | 0.151 | 0.201 | 0.253 | 0.322 | 0.084 | 0.183 | 0.337 | 0.791 |
| | Ours | 0.137 | 0.146 | 0.16 | 0.183 | 0.151 | 0.197 | 0.247 | 0.318 | 0.079 | 0.172 | 0.301 | 0.765 |
| PathFormer | original paper | 0.145 | 0.167 | 0.186 | 0.231 | 0.156 | 0.206 | 0.254 | 0.340 | - | - | - | - |
| | our reproduce | 0.155 | 0.163 | 0.183 | 0.212 | 0.159 | 0.206 | 0.255 | 0.341 | 0.098 | 0.199 | 0.36 | 0.72 |
| | FreqMask | 0.151 | 0.163 | 0.181 | 0.206 | 0.160 | 0.210 | 0.254 | 0.339 | 0.096 | 0.207 | 0.398 | 0.745 |
| | FreqMix | 0.152 | 0.161 | 0.181 | 0.210 | 0.161 | 0.207 | 0.257 | 0.340 | 0.099 | 0.219 | 0.356 | 0.773 |
| | Ours | 0.15 | 0.159 | 0.178 | 0.204 | 0.159 | 0.205 | 0.252 | 0.337 | 0.091 | 0.179 | 0.342 | 0.69 |
| PDF | original paper | 0.127 | 0.145 | 0.162 | 0.200 | 0.147 | 0.192 | 0.244 | 0.318 | - | - | - | - |
| | our reproduce | 0.128 | 0.146 | 0.161 | 0.197 | 0.147 | 0.192 | 0.244 | 0.320 | 0.082 | 0.173 | 0.325 | 0.830 |
| | FreqMask | 0.127 | 0.143 | 0.164 | 0.195 | 0.150 | 0.194 | 0.244 | 0.320 | 0.085 | 0.189 | 0.337 | 0.857 |
| | FreqMix | 0.129 | 0.144 | 0.161 | 0.192 | 0.149 | 0.196 | 0.247 | 0.322 | 0.082 | 0.176 | 0.327 | 0.839 |
| | Ours | 0.126 | 0.143 | 0.158 | 0.182 | 0.147 | 0.191 | 0.242 | 0.318 | 0.080 | 0.169 | 0.315 | 0.826 |

(a) Long-term prediction    (b) Long-term prediction

(c) Short-term prediction    (d) Short-term prediction

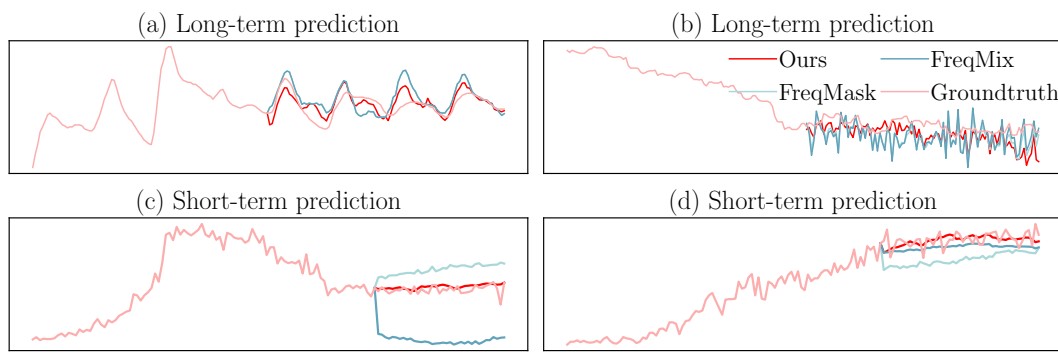

Figure 8: Example predictions of different methods under long-term (top) and short-term (bottom) protocols.

| Hypermeter | ETTh1 | | | | ETTh2 | | | | ETTm1 | | | | ETTm2 | | | |
|---|---|---|---|---|---|---|---|---|---|---|---|---|---|---|---|---|
| | 96 | 192 | 336 | 720 | 96 | 192 | 336 | 720 | 96 | 192 | 336 | 720 | 96 | 192 | 336 | 720 |
| Optimal $k$ | 4 | 4 | 4 | 4 | 2 | 2 | 2 | 4 | 3 | 3 | 2 | 2 | 4 | 4 | 2 | 4 |

Table 12: The optimal $k$ on ETT datasets using the iTransformer Liu et al. (2024) model.

| Hypermeter | Electricity | | | | Traffic | | | | Weather | | | | Exchange Rate | | | |
|---|---|---|---|---|---|---|---|---|---|---|---|---|---|---|---|---|
| | 96 | 192 | 336 | 720 | 96 | 192 | 336 | 720 | 96 | 192 | 336 | 720 | 96 | 192 | 336 | 720 |
| Optimal $k$ | 2 | 3 | 2 | 2 | 2 | 2 | 2 | 2 | 3 | 3 | 2 | 4 | 2 | 2 | 8 | 8 |

Table 13: The optimal $k$ on Electricity, Traffic, Weather, and Exchange Rate datasets using the iTransformer Liu et al. (2024) model.

## B.5 STANDARD DEVIATIONS

Tab. 14 to 17 shows the standard deviations of different runs, indicating the performance of our method is stable.

| Model | | ETTh1 | | | | ETTh2 | | | |
|---|---|---|---|---|---|---|---|---|---|
| | | 96 | 192 | 336 | 720 | 96 | 192 | 336 | 720 |
| iTrans former | Baseline | 0.392±0.001 | 0.447±0.002 | 0.483±0.003 | 0.516±0.003 | 0.303±0.001 | 0.381±0.000 | 0.412±0.001 | 0.434±0.002 |
| | Mask | 0.390±0.001 | 0.442±0.002 | 0.475±0.001 | 0.503±0.003 | 0.301±0.001 | 0.385±0.003 | 0.414±0.001 | 0.438±0.005 |
| | Mix | 0.388±0.002 | 0.440±0.002 | 0.477±0.000 | 0.504±0.004 | 0.301±0.001 | 0.380±0.001 | 0.414±0.001 | 0.434±0.003 |
| | Ours | **0.383±0.001** | **0.438±0.001** | **0.473±0.002** | **0.492±0.002** | **0.298±0.002** | **0.382±0.003** | 0.411±0.004 | **0.428±0.001** |

Table 14: Error bars on ETTh1 and ETTh2 datasets.

| Model | | ETTm1 | | | | ETTm2 | | | |
|---|---|---|---|---|---|---|---|---|---|
| | | 96 | 192 | 336 | 720 | 96 | 192 | 336 | 720 |
| iTrans former | Baseline | 0.344±0.002 | 0.383±0.003 | 0.421±0.001 | 0.494±0.003 | 0.183±0.001 | 0.251±0.002 | 0.311±0.001 | 0.412±0.001 |
| | Mask | 0.347±0.002 | 0.383±0.005 | 0.420±0.001 | 0.494±0.004 | 0.179±0.003 | 0.251±0.001 | 0.311±0.001 | 0.411±0.002 |
| | Mix | 0.334±0.005 | 0.375±0.002 | 0.421±0.000 | **0.485±0.002** | 0.178±0.002 | 0.248±0.001 | 0.311±0.000 | **0.407±0.002** |
| | Ours | **0.332±0.001** | **0.374±0.001** | 0.424±0.001 | 0.492±0.002 | **0.178±0.002** | **0.246±0.001** | **0.309±0.001** | **0.409±0.000** |

Table 15: Error bars on ETTm1 and ETTm2 datasets.

| Model | | Electricity | | | | Traffic | | | |
|---|---|---|---|---|---|---|---|---|---|
| | | 96 | 192 | 336 | 720 | 96 | 192 | 336 | 720 |
| iTrans former | Baseline | 0.152±0.000 | 0.159±0.001 | 0.179±0.003 | 0.230±0.013 | 0.399±0.001 | 0.418±0.000 | 0.428±0.000 | 0.463±0.000 |
| | Mask | 0.153±0.001 | 0.157±0.001 | 0.173±0.001 | 0.208±0.005 | 0.395±0.001 | **0.401±0.005** | **0.418±0.001** | 0.450±0.002 |
| | Mix | 0.151±0.000 | 0.158±0.001 | 0.173±0.000 | 0.205±0.003 | 0.400±0.003 | 0.414±0.004 | 0.424±0.002 | 0.453±0.003 |
| | Ours | **0.150±0.000** | **0.156±0.001** | **0.171±0.000** | **0.199±0.002** | **0.394±0.000** | 0.412±0.002 | 0.423±0.002 | **0.448±0.001** |

Table 16: Error bars on Electricity and Traffic datasets.

| Model | | Weather | | | | Exchange Rate | | | |
|---|---|---|---|---|---|---|---|---|---|
| | | 96 | 192 | 336 | 720 | 96 | 192 | 336 | 720 |
| iTrans former | Baseline | 0.175±0.001 | 0.224±0.001 | 0.281±0.000 | 0.362±0.003 | **0.086±0.000** | 0.180±0.000 | 0.335±0.002 | 0.856±0.004 |
| | Mask | 0.178±0.001 | 0.228±0.002 | 0.284±0.002 | 0.359±0.001 | 0.090±0.002 | 0.178±0.001 | 0.329±0.006 | 0.845±0.008 |
| | Mix | 0.175±0.001 | 0.224±0.000 | 0.279±0.000 | 0.354±0.000 | 0.089±0.001 | 0.178±0.001 | 0.328±0.006 | 0.868±0.008 |
| | Ours | **0.171±0.001** | **0.221±0.000** | **0.276±0.000** | **0.351±0.002** | **0.086±0.001** | **0.176±0.001** | **0.313±0.006** | **0.821±0.003** |

Table 17: Error bars on Weather and Exchange Rate datasets.

## C FEW-SHOT AND COLD-START

We tested our method on two settings where models are trained with small training set: the few-shot prediction Jin et al. (2024) and the cold-start Chen et al. (2023a). We strictly follow the settings of the original papers in our experiments. The results in Tab. 18 and 19 demonstrate that our method consistently outperforms FrAug with limited training samples.

### C.1 FEW-SHOT PREDICTION

Tab. 18 shows the results under few-shot prediction setting. We strictly follow the settings of the original paper Jin et al. (2024) in the comparisons. 100% data means training on the complete dataset, while 10% data means using 10% of the dataset to train the model. For FrAug, we use FreqMix and FreqMask to expand the dataset, selecting the best results to represent FrAug.

### C.2 COLD START

Tab. 19 demonstrates the Cold-start with few training samples (1%) available.

Table 18: Mean Squared Error (MSE) of the Few-shot long-term prediction on the ETTh1, ETTh2, Electrici datasets using PatchTST and iTransformer.

| | Method | ETTh1 | | | | ETTh2 | | | | Eletricity | | | |
|---|---|---|---|---|---|---|---|---|---|---|---|---|---|
| | | 96 | 192 | 336 | 720 | 96 | 192 | 336 | 720 | 96 | 192 | 336 | 720 |
| Patch | 100% data | 0.374 | 0.416 | 0.428 | 0.452 | 0.278 | 0.337 | 0.382 | 0.382 | 0.130 | 0.148 | 0.163 | 0.197 |
| | 10% data | 0.516 | 0.598 | 0.657 | 0.765 | 0.353 | 0.403 | 0.426 | 0.477 | 0.140 | 0.160 | 0.180 | 0.241 |
| | FrAug | 0.439 | 0.541 | 0.626 | 0.623 | 0.329 | 0.382 | 0.413 | 0.466 | 0.138 | 0.159 | 0.176 | 0.211 |
| | Ours | 0.420 | 0.476 | 0.543 | 0.601 | 0.322 | 0.382 | 0.401 | 0.453 | 0.136 | 0.159 | 0.174 | 0.204 |
| iTrans | 100% data | 0.392 | 0.447 | 0.483 | 0.516 | 0.303 | 0.381 | 0.412 | 0.434 | 0.152 | 0.159 | 0.179 | 0.230 |
| | 10% data | 0.668 | 0.730 | 0.815 | 0.915 | 0.348 | 0.443 | 0.485 | 0.501 | 0.203 | 0.203 | 0.226 | 0.269 |
| | FrAug | 0.601 | 0.662 | 0.761 | 0.906 | 0.334 | 0.432 | 0.486 | 0.501 | 0.189 | 0.190 | 0.211 | 0.248 |
| | Ours | 0.582 | 0.659 | 0.742 | 0.883 | 0.332 | 0.430 | 0.483 | 0.502 | 0.188 | 0.190 | 0.211 | 0.246 |

Table 19: Mean Squared Error (MSE) of the coldstart long-term prediction on the ETTh1, ETTh2, Electricity datasets using PatchTST and iTransformer.

| | Method | ETTh1 | | | | ETTh2 | | | | Eletricity | | | |
|---|---|---|---|---|---|---|---|---|---|---|---|---|---|
| | | 96 | 192 | 336 | 720 | 96 | 192 | 336 | 720 | 96 | 192 | 336 | 720 |
| Patch | 100% data | 0.374 | 0.416 | 0.428 | 0.452 | 0.278 | 0.337 | 0.382 | 0.382 | 0.130 | 0.148 | 0.163 | 0.197 |
| | 1% data | 0.564 | 0.660 | 0.624 | 0.782 | 0.402 | 0.512 | 0.479 | 0.503 | 0.169 | 0.203 | 0.216 | 0.287 |
| | FrAug | 0.479 | 0.567 | 0.597 | 0.692 | 0.372 | 0.481 | 0.437 | 0.486 | 0.153 | 0.179 | 0.196 | 0.266 |
| | Ours | 0.446 | 0.492 | 0.588 | 0.633 | 0.361 | 0.462 | 0.412 | 0.475 | 0.150 | 0.169 | 0.182 | 0.240 |
| iTrans | 100% data | 0.392 | 0.447 | 0.483 | 0.516 | 0.303 | 0.381 | 0.412 | 0.434 | 0.152 | 0.159 | 0.179 | 0.230 |
| | 1% data | 0.732 | 0.762 | 0.799 | 0.816 | 0.379 | 0.481 | 0.512 | 0.533 | 0.264 | 0.254 | 0.267 | 0.302 |
| | FrAug | 0.651 | 0.679 | 0.732 | 0.765 | 0.356 | 0.462 | 0.503 | 0.501 | 0.223 | 0.234 | 0.275 | 0.278 |
| | Ours | 0.627 | 0.663 | 0.720 | 0.732 | 0.351 | 0.450 | 0.499 | 0.480 | 0.219 | 0.225 | 0.246 | 0.289 |

