# OpenReview forum: "Dominant Shuffle: An Incredibly Simple but Exceptionally Effective Data Augmentation Method for Time-Series Prediction"
_ICLR.cc/2025/Conference — Submitted to ICLR 2025_

### Official Review · Reviewer_jp2H · 2024-10-29

**Soundness:** 2
**Presentation:** 2
**Contribution:** 2
**Rating:** 3
**Confidence:** 4

**Summary:**

This paper introduces "Dominant Shuffle," a novel data augmentation technique for time-series prediction, emphasizing a frequency-domain approach.

**Strengths:**

1. The proposed Dominant Shuffle is straightforward and easy to use.
2. The paper is easy to follow.

**Weaknesses:**

1. Although the proposed method appears simple yet effective, the authors should rigorously conduct and present experimental results. Currently, the reported experimental outcomes are inconsistent with those in the baseline papers. For instance, all accuracy results for baselines like iTransformer are weaker than those originally reported. In some cases, the augmented results even underperform compared to the original models. Specifically for SciNet, the baseline results show significant discrepancies from those presented in the original baseline paper when using horizons of 336 and 720 on the ETTH2 dataset.
2. This method focuses only on dominant frequencies, potentially overlooking minor or lower-magnitude frequencies that could contain valuable information. In certain time-series data, these less prominent frequencies may carry essential signals.
3. For time-series data with evolving frequency characteristics over time, like electricity load or financial series, a static frequency shuffling approach might not capture these dynamic changes effectively.

**Questions:**

See weaknesses

---

### Official Review · Reviewer_zq1L · 2024-11-01

**Soundness:** 2
**Presentation:** 2
**Contribution:** 2
**Rating:** 5
**Confidence:** 5

**Summary:**

The paper introduces a data augmentation technique, "dominant shuffle," designed for time-series prediction. The approach aims to address limitations in current frequency-domain augmentations, which may produce out-of-distribution samples that negatively affect model performance. By limiting augmentations to dominant frequencies and using a shuffle method to avoid adding noise, dominant shuffle seeks to preserve the underlying data structure while enhancing prediction accuracy. The authors validate the method through experiments across various datasets and models, demonstrating consistent improvements. However, the approach is heuristic-based, lacking a solid theoretical foundation, which the authors acknowledge as an area for future exploration.

**Strengths:**

1. Unlike in the field of computer vision, real-world data for time-series analysis is quite limited, and data augmentation methods for time-series data are still not as well-developed as in CV. This is an important area of research.

2. The experimental results suggest that dominant shuffle consistently improves the performance of existing models.

3. Figure 2 appears to indicate that out-of-distribution (OOD) samples generated by data augmentation significantly impact model performance, which is a direction worth further investigation.

**Weaknesses:**

1. The theoretical section of this paper lacks a solid and in-depth justification as to why dominant shuffle performs better. Figure 4 seems to attempt to illustrate this, but dominant shuffle clearly changes the periodicity of the data. Why this change should be better than simply adding minor noise or removing dominant frequencies is still not convincingly explained.

2. As a reviewer, I accessed the publicly available code for this paper during its submission to NeurIPS and attempted to replicate the results. Our findings indicate that, while dominant shuffle can occasionally enhance model performance, this improvement is inconsistent and sometimes even causes a performance drop. It appears the method is still influenced by randomness, suggesting it does not consistently enhance model outcomes.

3. Based on point 2, I believe it would be more appropriate for the experiments to provide average results over five to ten trials for greater reliability.

**Questions:**

Q1. See weakness 3.

Q2. Could the authors provide quantitative metrics to analyze the severity of OOD samples generated by different DA methods?

---

### Official Review · Reviewer_hwXn · 2024-11-04

**Soundness:** 2
**Presentation:** 3
**Contribution:** 2
**Rating:** 5
**Confidence:** 4

**Summary:**

This paper proposes Dominant Shuffle, a simple yet highly effective data augmentation technique for time series prediction. The method addresses the domain gap between augmented and original data by restricting perturbations to dominant frequencies and employing shuffling to minimize the impact of external noise. While this approach is straightforward and yields positive results, it is primarily heuristic in nature and lacks a robust theoretical foundation. It demonstrates good performance when integrating with existing backbone networks.

**Strengths:**

- The paper is well-written and easy to follow.
- The paper presents an interesting idea to augment time series data in the frequency domain.
- The proposed augmentation can improve the performance of existing approaches on the forecasting tasks.

**Weaknesses:**

- The proposed method lacks robust theoretical justification, resembling a heuristic approach rather than a rigorously validated technique. How to determine the magnitude of modification over the frequencies? Is it a hyperparameter?

- While the proposed method is both simple and effective for models without augmentation, its performance improvement over existing augmentation methods appears minimal, particularly as illustrated in Figure 2. If the results are so closely aligned, what is the primary advantage of adopting the proposed method?

- The experiments appear to focus solely on prediction tasks, overlooking other significant areas such as imputation and anomaly detection, which are also widely recognized in time series analysis.

**Questions:**

- Is there a more theoretical proof available for the proposed data augmentation method?

- Could you clarify the limited improvements observed over existing augmentation methods? If the performance is so similar, what is the primary advantage of utilizing the proposed method?

- I understand that it may be challenging during the rebuttal phase, but if possible, could you provide some preliminary results on additional tasks such as imputation or anomaly detection? Thanks.

- How to determine the magnitude of modification over the frequencies? Is it a hyperparameter?

---

### Official Review · Reviewer_Fqeo · 2024-11-04

**Soundness:** 3
**Presentation:** 3
**Contribution:** 2
**Rating:** 6
**Confidence:** 5

**Summary:**

The paper introduces a new data augmentation technique called "Dominant Shuffle" for time-series prediction tasks. The authors argue that existing frequency-domain data augmentation methods introduce excessive variability, leading to out-of-distribution samples that can harm model performance. To address this, they propose using a shuffling approach on dominant frequencies to preserve the original structure while avoiding external noise. A lot of experiments are conducted to evaluate the proposed method, which is appreciated.

**Strengths:**

1. The dominant shuffle method is simple yet effective. And the paper demonstrates that dominant shuffle outperforms existing data augmentation techniques, including those that are more complex.
2. The method consistently improves performance across various state-of-the-art models and diverse datasets, indicating its robustness and generalizability.

**Weaknesses:**

1. While the paper emphasizes the issue of data scarcity in Introduction, it will be better to include experiments on such scenario. Different augmentation sizes are analyzed but different original data sizes are not analyzed.
2. I appreciate the work made by authors, but this paper provides limited insights to the community. Perturbing top-k frequencies with highest magnitudes is the existing technic. Although dominant shuffle leads to performance improvement, it is more like a trick, with limited novelty.

**Questions:**

Why does simply shuffling the dominant frequencies result in significant performance improvements? Is there any theoretical basis or empirical evidence that supports the effectiveness of this approach?

---

### Meta-Review · Area_Chair_skUG · 2024-12-14

**Metareview:**

**(a) Summary**

This paper addresses the problem of frequency-domain data augmentation for time series prediction. The proposed method involves shuffling the top-k components of a given time series in the frequency domain and reconstructing it to generate new time series data. The approach is empirically evaluated using real-world time series datasets.

**(b) Strengths**

- **Relevance:** Data augmentation for time series analysis is an important and relevant research topic, as the amount of time series data is often insufficient to train modern machine learning models.
- **Simple approach:** The proposed method is straightforward and easy to implement. If proven effective, it could provide valuable insights.

**(c) Weaknesses**

- **Novelty:** The technical contribution of the paper is limited, as the proposed method relies on straightforward heuristic approaches.
- **Quality:** The effectiveness of the proposed method is not convincingly demonstrated. While simplicity can be an advantage, the lack of theoretical justification or rigorous validation significantly weakens the contribution, as also noted by the reviewers.
- **Empirical evaluation:** As highlighted by **Reviewer zq1L**, the performance improvements reported in the experiments are unconvincing, likely due to the randomness inherent in the proposed method. This issue requires careful examination through multiple trials to ensure robustness and reliability.

**(d) Decision Reasoning**

The weaknesses outlined above constitute significant limitations. Most critically, the lack of rigor in the methodological contribution and its validation justifies a recommendation for rejection.

**Additional Comments On Reviewer Discussion:**

No rebuttal was provided, and reviewers recommended rejecting the paper.

---

### Decision · Program_Chairs · 2025-01-22

Reject